

# Projecting sediment export from two highly glacierized alpine catchments under climate change: Exploring non-parametric regression as an analysis tool

Lena Katharina Schmidt[1], Till Francke[1], Peter Martin Grosse[1], Axel Bronstert[1]

[1]Institute of Environmental Sciences and Geography, University of Potsdam, Potsdam, 14476, Germany

*Correspondence to*: L. Katharina Schmidt (leschmid@uni-potsdam.de)

**Abstract.** Future changes in suspended sediment export from deglaciating high-alpine catchments affect
downstream hydropower and reservoirs, flood hazard, ecosystems and water quality. Yet so far, quantitative projections of future sediment export have been hindered by the lack of physical models that can take into account all relevant processes within the complex systems determining sediment dynamics at the catchment scale. As a promising alternative, machine-learning (ML) approaches have recently been successfully applied to modeling suspended sediment yields (SSY).

This study is the first to our knowledge exploring machine-learning approach to derive sediment export projections until the year 2100. We employ Quantile Regression Forest (QRF), which proved to be a powerful method to model past SSY in previous studies, at two nested high-alpine gauges in the Ötztal, Austria, i.e. gauge Vent (98.1 km² catchment area, 28 % glacier cover in 2015) and gauge Vernagt (11.4 km² catchment area, 64 % glacier cover). As predictors, we use temperature and precipitation projections (EURO-CORDEX) and discharge projections
(AMUNDSEN physically-based hydroclimatological and snow model) for the two gauges. We address uncertainties associated with a known limitation of QRF, i.e. that underestimates can be expected if out-of-observation-range (OOOR) data points (i.e. values exceeding the range represented in the training data) occur in the application period. For this, we assess the frequency and extent of these exceedances and the sensitivity of the resulting mean annual suspended sediment concentration (SSC) estimates. We examine the resulting SSY
projections for trends, the estimated timing of 'peak sediment' and changes in the seasonal distribution.

Our results show that the uncertainties associated with the OOOR data points are small before 2070 (max. 3 % change in estimated mean annual SSC). Results after 2070 have to be treated more cautiously, as OOOR data points occur more frequently and as glaciers are projected to have (nearly) vanished by then in some projections, which likely substantially alters sediment dynamics in the area. The resulting projections suggest decreasing
sediment export at both gauges in the coming decades, regardless of the emission scenario, which implies that 'peak sediment' has already passed or is underway. Nevertheless, high(er) annual yields can occur in response to heavy summer precipitation, and both developments would need to be considered in managing sediments as well as e.g. flood hazard. While we chose the predictors to act as proxies for sediment-relevant processes, future studies are encouraged to try and include geomorphological changes more explicitly, e.g. changes in connectivity,
landsliding / rockfalls, or vegetation colonization, as these could improve the reliability of the projections.



## 1 Introduction

Fluvial suspended sediment export from glacierized, high-alpine areas can be up to a magnitude higher (per unit area) than in non-glacierized downstream areas (Hinderer et al., 2013; Beniston et al., 2018). Thus, sediment dynamics in these high-alpine areas and changes therein have important implications for downstream hydropower generation and reservoir sedimentation (Schöber and Hofer, 2018; Guillén-Ludeña et al., 2018; Li et al., 2022), water quality (as well as nutrient and contaminant transport) (Bilotta and Brazier, 2008), aquatic species and riverine ecosystems (Milner et al., 2009, 2017; Gabbud and Lane, 2016; Huss et al., 2017), but also flood hazard (Nones, 2019) and carbon cycling (Tan et al., 2017; Syvitski et al., 2022).

High-alpine areas are particularly sensitive to climate change, experience above-average warming (Gobiet et al., 2014) and hence crucial cryospheric changes, such as ongoing and accelerating deglaciation, permafrost melt and snow cover changes (Huss et al., 2017; Beniston et al., 2018; Abermann et al., 2009). These changes go hand in hand with changes in discharge volumes, timing and magnitude (Vormoor et al., 2015; Kuhn et al., 2016; van Tiel et al., 2019; Rottler et al., 2020; Hanus et al., 2021). This in turn affects sediment export, and past changes have been observed frequently, e.g. due to enhanced subglacial sediment evacuation and increased sediment accessibility in expanding erodible landscapes (Micheletti and Lane, 2016; Carrivick and Heckmann, 2017; Lane et al., 2017, 2019; Costa et al., 2018; Delaney and Adhikari, 2020; Li et al., 2020; Vergara et al., 2022).

Nevertheless, future changes in sediment export are understudied (Zhang et al., 2022) and questions such as "Are sediment yields from deglaciating catchments increasing, decreasing or is there no pattern?" or "to what extent is it possible to quantify spatio-temporal patterns of future sediment yields?" (Carrivick and Tweed, 2021) have yet to be answered – although projections of climatological (e.g. Gobiet and Kotlarski, 2020; Gobiet et al., 2014) , glaciological (e.g. Stoll et al., 2020; Bolibar et al., 2022; Huss, 2011) and hydrological changes (e.g. Madsen et al., 2014; Hanzer et al., 2018; Hanus et al., 2021; Huss and Hock, 2018; Tecklenburg et al., 2012; Wijngaard et al., 2016), that could serve as a basis for estimating future changes in sediment export, are numerous.

The main reason why answering such questions is challenging is that modeling sediment export at the catchment scale with process-based models remains difficult – if not impossible – because it is determined by a complex system of interconnected processes that is not straightforward to capture. For example, the relationship between suspended sediment concentrations and discharge is most often nonlinear in time and space, and univariate models relying solely on discharge are often insufficient (Vercruysse et al., 2017; Zhang et al., 2021). Hence, in addition to variations in discharge, changes in sediment availability, entrainment, transport and deposition would have to be considered, there may be threshold effects and nonlinear responses of geomorphic processes (e.g. triggering of mass movements or debris flows), correlated influencing factors, hysteresis and seasonality (Huggel et al., 2012; Landers and Sturm, 2013; Vercruysse et al., 2017; Costa et al., 2018; Schmidt et al., 2022; Zhang et al., 2022). Additionally, long-term field observations (i.e. several decades and covering a wide range of conditions) that provide enough training and validation data to develop sediment-yield models or to analyze trends are very rare (Zhang et al., 2022; Schmidt et al., 2023).

There are conceptual models on (suspended) sediment export from deglaciating areas (Antoniazza and Lane, 2021; Carrivick and Tweed, 2021; Zhang et al., 2022), which expect an initial increase in sediment export as glaciers begin to retreat, and an eventual decrease – after 'peak sediment' – once the glaciers have disappeared and the landscape stabilizes. The timing of 'peak sediment' is presumed to depend i.a. on changes in erosive precipitation,





i.e. a negative trend in erosive precipitation implies that peak meltwater and peak sediment may co-occur, while a positive trend or no change in erosive precipitation result in a lag between peak meltwater and peak sediment. However, deducing estimates of future sediment export and implications for individual catchments based on these conceptual considerations is not straightforward or even possible.

Common approaches to model sediment yields at the catchment scale, such as SWAT (e.g. Vigiak et al., 2017),
BQART (Syvitski and Milliman, 2007), WBMsed (Cohen et al., 2013), WASA-SED (Mueller et al., 2010) or SAT (Zhang et al., 2021), are mostly empirical or conceptual in their sediment modules, do not consider all relevant erosion processes (i.e. neglecting glacial, gully erosion and landslides in the case of SWAT) and often concentrate on large spatial scales (i.e. sediment fluxes to the oceans for large basins, entire continents or at global scale) and / or large temporal scales (i.e. multiyear averages and long-term fluxes). On the other end of the spectrum, models
for individual parts or processes within glacierized catchments exist, as for example a numerical approach to model subglacial fluvial sediment transport (Delaney et al., 2019) that has also been coupled with models for ice dynamics and bedrock erosion (Delaney et al., 2021), or e.g. probabilistic or physical models of mass wasting processes, such as landslide or debris flows (Iverson and George, 2014; Hirschberg et al., 2021; Campforts et al., 2022). However, as of yet, there is no all-in-one physical model (fully-distributed, incorporating thermal and pluvial
drivers of sediment mobilization and transport) to simulate sediment export from cryospheric basins (Zhang et al., 2022) at the catchment scale.

Accordingly, studies that have attempted to project future suspended sediment yields (SSY) chose rather qualitative approaches, such as comparing sediment yield observations of warmer and colder ablation seasons (Stott and Mount, 2007; Bogen, 2008), using responses of SSY to past predictor changes and applying this to
projected changes in the future (Li et al., 2021b) or fitting a multiple regression model to past data (of only one year) and increasing the temperature input in the model (Stott and Convey, 2021). However, these approaches may preclude modeling decreases or accounting for interactions between variables.

As a promising alternative, geoscientific machine-learning approaches have emerged, and have recently been acknowledged for their potential in applications to Earth System Science (Reichstein et al., 2019). Indeed, first
studies showed that machine-learning approaches can easily outperform well-known existing models for sediment yield (Gupta et al., 2021; Rahman et al., 2022; Jimeno-Sáez et al., 2022; Schmidt et al., 2023). In a previous study, we have developed and validated a Quantile Regression Forest (QRF) approach to model SSY in two nested high-alpine catchments and estimate yields for the past five decades (Schmidt et al., 2023). This showed that the QRF model outperformed commonly applied sediment rating curves by about 20 % of explained variance, and other
studies found that regression trees and Random Forest models (which QRF is based on) even outperformed other machine learning approaches in modeling sediment dynamics (Talebi et al., 2017; Al-Mukhtar, 2019).

Thus, the present study is motivated to explore QRF to model future SSY based on measurement data, emission scenarios and subsequent hydrological model results. We test the approach in two glacierized high-alpine catchments in the Ötztal in Austria, where projections of future climatological and glacio-hydrological conditions
from the AMUNDSEN model are available (Hanzer et al., 2018), and where we have successfully trained and applied QRF models to reconstruct past sediment export, using records of discharge, precipitation and air temperature (Schmidt et al., 2023).



The goals of the present study are (i) to derive estimates of future changes in sediment export with respect to trends in annual yields, shifts in the seasonal distribution and the timing of 'peak sediment' and (ii) to assess uncertainties of the model due to known limitations of the QRF method in order to identify the limitations of the approach.

## 2    Methods

In a previous study, we trained and validated quantile regression forest models to retrospectively estimate SSY at two gauges for the past 5 decades, using the available records of turbidity-derived suspended sediment concentrations (four and 15 years) and long-term records of the predictors, i.e. discharge, precipitation and temperature (Schmidt et al., 2023) (Figure 2, dashed-line box). In the present study, we use these models and apply them to downscaled and bias-corrected EURO-CORDEX temperature and precipitation projections that were used as input data for the glacio-hydrological model AMUNDSEN as well as the discharge projections of AMUNDSEN (Hanzer et al., 2018)(Figure 1). In the following, we outline the Quantile Regression Forest approach including its advantages and limitations with respect to modeling suspended sediment dynamics and the choice of predictors to model sediment dynamics in high-alpine areas. Then, we describe the study area, input data and necessary adjustments, as well as how we analyzed the limitations, sensitivities and the resulting SSY estimates.

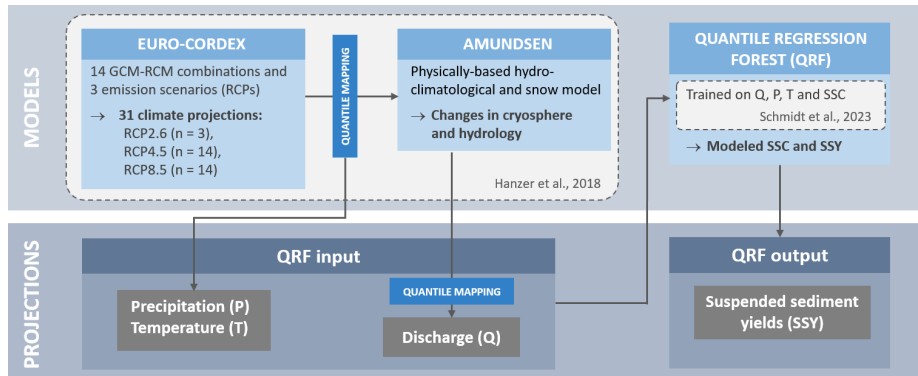

*Figure 1 Overview of models and resulting projections used in this study. Bias-corrected EURO-CORDEX climate projections and AMUNDSEN model results serve as input data for the QRF models. Q: discharge, P: precipitation, T: temperature, SSC: suspended sediment concentrations, SSY: suspended sediment yields.*

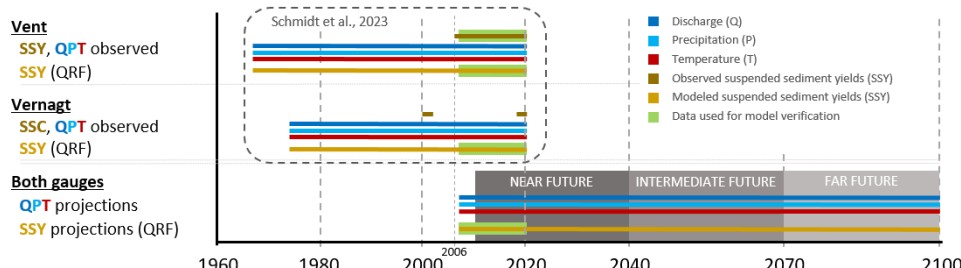

*Figure 2 Temporal extent of input data as well as modeled suspended sediment yields (SSY) in the previous study (dashed-line box, topleft), as well as projections (discharge (Q), precipitation (P), temperature (T)) as input data and SSY estimates from this study. 2006 – 2020 is the training data period at gauge Vent. At gauge Vernagt, the QRF model was trained on the years 2000, 2001, 2019 and 2020, when SSC data were available. Thus, to verify model results in the present study, we use the QRF estimated yields at gauge Vernagt of the years overlapping with the climate and hydrology projections, i.e. the "overlap period" 2007 – 2020 (section 2.4.1).*



## 2.1 Quantile Regression Forest for suspended sediment concentration modelling

Quantile Regression Forest (QRF) (Meinshausen, 2006) is a non-parametric regression technique, that is based on Random Forest (RF) and can be classified as a machine-learning approach. It learns from the training data by growing ensembles of regression trees on random subsets (bootstrap samples) of the training data (Francke et al., 2008a; Schmidt et al., 2023). In each regression tree, the data are recursively partitioned based on splitting rules, where both RF and QRF randomly select the predictors used for splitting. In contrast to Random Forest, QRF keeps all observations within a node (whereas RF only keeps the mean), which allows to construct prediction intervals and to assess uncertainty (ibid.).

The advantages of QRF include that it can handle multiple input variables, makes no assumptions on distributions and can deal with interactions, non-linearity and non-additive behavior. As limitations, it does not allow for easy interpretation of effects of single predictors and model predictions will always be within the range of observations, i.e. if the predictors in the period of application exceed the range represented in the training dataset (hereafter called "out-of-observation-range (OOOR) data points"), we can expect over- (or under-) estimations (ibid.) of the target variable, if the respective predictor has a continuing monotonic effect in this range.

With respect to modelling suspended sediment concentrations, studies have shown that QRF is very well-suited to model sedigraphs and estimate annual SSY (Francke et al., 2008b, a; Zimmermann et al., 2012) and that it performs favourably compared to sediment rating curves and generalized linear models (Francke et al., 2008a; Schmidt et al., 2023). On a related note, RF (which QRF is based on) outperformed support-vector machines and artificial neural networks (Al-Mukhtar, 2019) in modelling suspended sediment concentrations.

In a previous study (Schmidt et al., 2023), we trained QRF models on data of the two gauges Vent and Vernagt, using the limited available time series of turbidity (4 and 15 years) and long records of the primary predictors, discharge (Q), precipitation (P) and air temperature (T) (Figure 2). These can be seen as drivers or proxies for processes and catchment conditions crucial to sediment dynamics in high-alpine areas: e.g. discharge determines sediment transfer and erosion within the channel, precipitation is key for runoff formation and hillslope erosion, hillslope-channel coupling and the triggering of mass movement events, and air temperature controls the activation of sediment sources (e.g. sub- and proglacial sediments and their transport by glacier meltwaters or hillslope destabilization by permafrost thaw) and whether precipitation occurs as rain or snow. In addition to these primary predictors, we derived ancillary predictors to describe antecedent conditions and cumulative effects thereof: e.g. longer-term discharge behavior may exhaust sediment sources or lead to sediment storage, long warm periods may deplete snow cover and accelerate glacier melt associated with increased subglacial sediment transport, and high antecedent moisture conditions may amplify surface runoff or promote mass movements in response to precipitation events. The final models performed well and favorably to sediment rating curves, even with respect to threshold effects. For the past 5 decades, OOOR data points (see section 2.4.2) were rare, which strengthened the notion that the available training data covered the majority of typical situations.

## 2.2 Study area

The two studied gauges Vent Rofenache (hereafter "Vent", operated by the Hydrographic Service of Tyrol) and Vernagt (operated by the Bavarian Academy of Sciences and Humanities) are located in the Rofental in the Ötztal



Alps, Austria (Figure 3). The two corresponding nested catchments of 98.1 km² and 11.4 km² span elevations ranging from 1891 m a.s.l. at gauge Vent and 2635 m a.s.l. at gauge Vernagt to 3772 m a.s.l. The area is characterized by a relatively warm and dry climate (for this alpine setting), with average annual precipitation as low as 660 mm at gauge Vent but a strong precipitation gradient with elevation (Schmidt et al., 2023). Both catchments are heavily glacierized (28 % and 64 % glacier cover in 2015 (Buckel and Otto, 2018)), but accelerating glacier retreat has been observed since the beginning of the 1980s (Escher-Vetter and Siebers, 2007; Braun et al., 2007; Abermann et al., 2009). Apart from the glaciers, land cover at high elevations is dominated by bare rock or sparsely vegetated terrain, whereas mountain pastures and coniferous forests occupy lower elevations. Geology is dominated by biotite-plagioclase, biotite and muscovite gneisses, variable mica schists and gneissic schists (Strasser et al., 2018).

The river Rofenache is a tributary stream of the Ötztaler Ache, one of the largest tributaries to the river Inn. The glacial to nival hydrological regime shows a pronounced seasonality, with almost 90 % of discharge occurring during snow and glacier melt from April to September (Schmidt et al., 2022). Mean annual suspended sediment concentrations at gauge Vent were the highest in an Austria-wide comparison (Lalk et al., 2014). Annual suspended sediment yields in Vent averaged 1500 t km$^{-2}$ a$^{-1}$ with an even more pronounced seasonality compared to discharge (99 % of the annual SSY transported from April to September) (Schmidt et al., 2022).

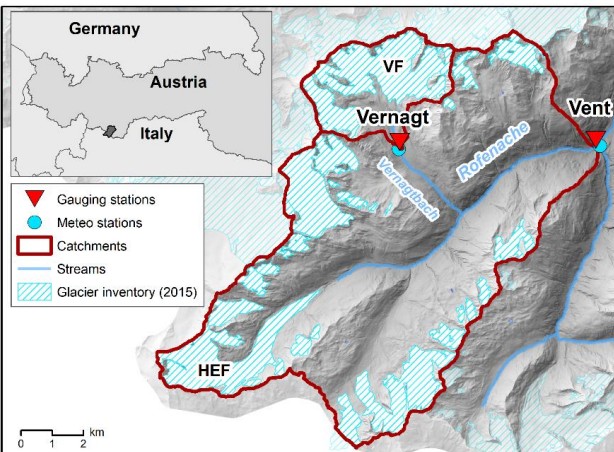

*Figure 3 Map of the catchment area above gauge Vent, with nested catchment above gauge Vernagt and major glaciers Vernagtferner (VF) and Hintereisferner (HEF). Meteo stations recording precipitation and temperature are located close to the gauges. (Map based on 10 m DEM of Tirol (Land Tirol, 2016), glacier inventory of 2015 (Buckel and Otto, 2018) and river network from tiris open government data (Land Tirol, 2021).)*

### 2.3 Input data

#### 2.3.1 Climate projections

We used projections of air temperature and precipitation of the European part of the COordinated Regional Downscaling Experiment (EURO-CORDEX) (Jacob et al., 2014), that have been downscaled and bias-corrected for use in their hydrological model by Hanzer et al., 2018. The EURO-CORDEX initiative provides regional climate model results to enable exploring impacts of future climate change at comparatively high horizontal resolution. This is beneficial for modelling future sediment export, for examples since regional climate model



simulations provide higher precipitation intensities, which are entirely missing in the global climate model
simulations (Jacob et al., 2014), and are thus more likely to capture erosion-relevant changes in precipitation. The
data used in this study and by Hanzer et al., 2018, were the result of six different regional climate models (RCMs)
driven by five different global climate models (GCMs), resulting in a total of 14 different GCM-RCM modeling
chains (Table 1). These are forced by three different emission scenarios expressed as representative concentration
pathways (RCP), which correspond to an added radiative forcing of 2.6, 4.5 and 8.5 W/m² at the end of the 21$^{st}$
century relative to pre-industrial conditions, i.e. RCP2.6 (intervention scenario assuming peak $CO_2$ concentrations
in the middle of the century, followed by slow decline and negative emissions), RCP4.5 (intermediate scenario
with peak emissions mid-century followed by strong decline) and RCP8.5 (assuming no implementation of climate
mitigation policies, considerably and steadily increasing emissions and greenhouse gas concentrations over time)
(Jacob et al., 2014; Hanzer et al., 2018). This results in a total of 31 RCP-GCM-RCM combinations. The horizontal
resolution of the original EURO-CORDEX projections is 0.11° (≈ 12.5 km).

Hanzer et al. used statistical downscaling to represent the local scale, i.e. quantile mapping, to match the
distributions of the climate model simulations of the current climate to the distributions of observations from
stations. This is necessary, especially in Alpine regions, because the 12.5 km spatial resolution of RCMs (despite
being comparatively high) can not sufficiently resolve topographical and climatological heterogeneities
(Hirschberg et al., 2021). Hanzer et al. interpolated the meteorology to a 100 m grid and we used the projections
for the two grid cells that are located closest to the gauges Vent and Vernagt. EURO-CORDEX simulations are
provided at daily resolution, and Hanzer et al., have disaggregated them to 3 h resolution to capture diurnal
variability in the energy fluxes. We re-aggregated these data to daily resolution to match the temporal resolution
of our QRF models (see section 2.3.3).

*Table 1 Overview of EURO-CORDEX scenario simulations used in this study (unaltered from Hanzer et al. (2018), distributed
under CC BY 3.0 https://creativecommons.org/licenses/by/3.0/).*

| ID | RCM | GCM | RCPs |
|----|-----|-----|------|
| 1 | CCLM4-8-17 | CNRM-CM5 | 4.5, 8.5 |
| 2 | CCLM4-8-17 | EC-EARTH | 4.5, 8.5 |
| 3 | CCLM4-8-17 | HadGEM2-ES | 4.5, 8.5 |
| 4 | CCLM4-8-17 | MPI-ESM-LR | 4.5, 8.5 |
| 5 | HIRHAM5 | EC-EARTH | 2.6, 4.5, 8.5 |
| 6 | RACMO22E | EC-EARTH | 4.5, 8.5 |
| 7 | RACMO22E | HadGEM2-ES | 4.5, 8.5 |
| 8 | RCA4 | CNRM-CM5 | 4.5, 8.5 |
| 9 | RCA4 | EC-EARTH | 2.6, 4.5, 8.5 |
| 10 | RCA4 | CM5A-MR | 4.5, 8.5 |
| 11 | RCA4 | HadGEM2-ES | 4.5, 8.5 |
| 12 | RCA4 | MPI-ESM-LR | 4.5, 8.5 |
| 13 | REMO2009 | MPI-ESM-LR | 2.6, 4.5, 8.5 |
| 14 | WRF331F | CM5A-MR | 4.5, 8.5 |

### 2.3.2   Hydrological projections

We used discharge projections of the physically-based hydroclimatological and snow model AMUNDSEN
(Hanzer et al., 2018), which is a fully distributed energy and mass balance model including glacier evolution ($\Delta h$
method) and particularly adapted to high mountain catchments of small to regional scale. It comprises a glacier



retreat module and has been extensively validated for historic conditions, especially with respect to snow distribution (Hanzer et al., 2016). This is especially beneficial for modeling sediment dynamics, since AMUNDSEN can model processes such as changes in glacier melt that govern discharge dynamics and are crucial
to sediment fluxes in these high-alpine areas. AMUNDSEN was forced by the downscaled, bias-corrected and temporally disaggregated EURO-CORDEX simulations of precipitation and air temperature described above (as well as relative humidity, global radiation and wind speed), and modeled snow, glaciers and hydrology in the Ötztal Alps until 2100 (ibid.). Hanzer et al. have bias-corrected all RCM outputs using at least 20 years of observations in the period 1971 to 2005, and concluded that the corrected RCM outputs adequately represent the
mean and variability of the observed climate. The AMUNDSEN model was calibrated and extensively validated for the period 1997 – 2013, using water-balance-derived mean areal precipitation, snow depth recordings, Landsat and MODIS-derived snow extent maps, glacier mass balances and runoff recordings (Hanzer et al., 2016).

The temporal extent of both the meteorological and the hydrological projections is 2006 to 2100, but since data are not available for the entire year of 2006, we use the period of 2007 to 2100. Additionally, three HadGEM-
driven models ended in November 2099. The years 2007 to 2020 overlap with observation data at gauge Vent and results of the previous study at gauge Vernagt (Figure 2), which we utilize to verify our model results (see sections 2.4.1 and 3.2).

### 2.3.3   Adjustment of input data for the QRF model

As the QRF models were trained at daily resolution, we aggregated the Q and T projections from 3 h resolution to
daily means and P projections to daily sums. However, comparing the AMUNDSEN Q projections to observations in the overlap period (2007 – 2020; see Figure 2), showed that underestimation of Q during the glacier melt period at gauge Vernagt and substantial overestimation of Q during the snowmelt period at gauge Vent. Hanzer et al. (2018) have acknowledged that, but have left the overestimations (percent bias (PBIAS) of up to 23 %) unaltered, since "mainly changes than absolute values are analyzed; these partial biases likely do not affect the main
conclusions". However, in our case, it is necessary to correct the discharge data, since SSY are sensitive to discharge amounts and additionally, unrealistic discharge amounts exceeding the maximum discharge value in the training data represent a challenge (see section 2.4.2). Also, it is necessary to represent discharge seasonality, and thus discharge origins, as accurately as possible, as usually more sediment is exported during glacier melt than at similar discharge levels during snowmelt (Schmidt et al., 2022).

For consistency, we applied the same bias-correction as Hanzer et al., i.e. quantile mapping, using the methodology by (Gudmundsson et al., 2012) as implemented in the R package *qmap* (Gudmundsson, 2016). Due to strong season-dependent biases, Hanzer et al. have performed quantile mapping for each season individually. We followed this approach, yet in order to best represent discharge seasonality, we shifted the limits of the seasons by one month (NDJ, FMA, MJJ, ASO instead of DJF, MAM, JJA, SON), as this corresponded better to seasons with
similar characteristics of over- or underestimation.

### 2.4   Analyses of model limitations and uncertainties

To analyze model performance and identify the limits of the applicability of the presented QRF modeling approach, we verified the modeled SSY against measurement data in the overlap period (2007 – 2020) (section 2.4.1),




assessed the frequency of OOOR data points as well as by how much the observation range of the predictors is
exceeded in the projections and analyzed whether the modeled SSY are sensitive to changes in these predictors
(section 2.4.2).

### 2.4.1 Verification of model results based on observed data

To determine how well the SSY projections of our QRF models correspond to SSY derived from turbidity
measurements, we compared model results in the overlap period (2007 to 2020) to measurements at gauge Vent.
Lacking continuous direct measurements at gauge Vernagt, we used estimated SSY for the years 2007 - 2020 from
the QRF model trained on measurements for the years 2000, 2001, 2019 and 2020 (Schmidt et al., 2023) for the
comparison (see also Figure 2). To simplify the descriptions in the results, we hereafter refer to these estimates as
"observations" as well. As the hydroclimatic projections (and, thus, the SSY projections resulting from thereof)
do not mimic the characteristics of single years (let alone month or days), but merely reproduce their distribution,
we compared the distributions of observed and simulated annual SSY. To test for significant differences between
these distributions, we used the two-sample Kolmogorov-Smirnov test, which is able to handle the non-normal
distribution of some groups, as implemented in the R package *stats* version 3.5.1 (R Core Team, 2018).
Additionally, we assessed whether the seasonality of sediment export is accurately represented in the model results,
by comparing mean monthly SSY.

### 2.4.2 Assessment of limits of applicability

As mentioned in section 2.1, a known limitation of QRF is that model bias can potentially result if the predictors
in the application period exceed the range of observed values used as training data. This limitation is a direct
consequence of the numerical characteristics of RF and QRF, which are incapable of extrapolation. In order to
assess, how often and to what extent the model results are affected, we performed a series of analyses (overview
in Figure 4), described in the following.

**Analysis of out-of-observation-range days**

First, we quantified how often OOOR days occurred for each projection *proj* and predictor *p* (i.e. Q, P or T), as
the mean annual number of OOOR days per year:

$$\overline{n_{p,proj}} = \frac{1}{n_{years}} \cdot \sum_{i=1}^{n_{years}} n_{p,proj,i} ,$$ (1)

where $n_{years}$ is the number of years, and $n_{p,proj,i}$ is the number of OOOR days in a given predictor and projection in
a given year *i*. Additionally, we determined the **exceedance extent** (in % of the maximum value in the training
data $max(x_{p,train})$), i.e. by how much the maxima in the observations (i.e. the training period) $max(x_{p,train})$ were
exceeded on the OOOR days *j* (i.e. in the projection period),

$$e_{p,j} = \frac{x_{p,j} - max(x_{p,train})}{max(x_{p,train})} \cdot 100 \quad \forall \quad x_{p,j} > max(x_{p,train}),$$ (2)

where $x_{p,j}$ is the value of the predictor *p* on OOOR day *j*.



Although the same limitations of QRF theoretically also apply to predictors falling *below* the training minima, we did not consider them, since Q and P already contained very low or zero values and cannot fall below zero. Likewise, for T, minimum temperatures are already well below zero and further decrease (if it occurs) is not further physically relevant to sediment transport. Similarly, we only considered summer precipitation at both gauges (i.e.
May – September), to exclude snowfall events that are not directly relevant to sediment dynamics.

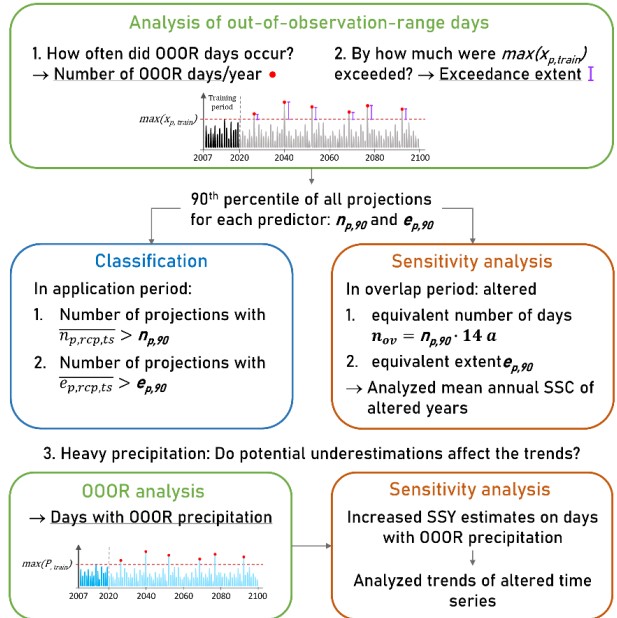

*Figure 4 Overview of analyses performed with respect to OOOR days. Max(xp,train) denotes the maximum value in the training data,* n *refers to the number of OOOR days and* e *to the exceedance extent. Subscripts:* p *for predictor,* rcp *for emission scenario,* ts *for time slice. The analysis of OOOR precipitation days (3.) is described in section 2.5.*

**Sensitivity analysis**

Second, we performed sensitivity analyses for the three primary predictors Q, P, and T (Figure 4), to assess the effects of the abovementioned exceedances on the model results. For this, we determined the 90th percentiles of the number of exceedances $\overline{n_{p,proj}}$, and the exceedance extents $e_{p,j}$ of all projections, i.e. $\boldsymbol{n_{p,90}}$ (in d a$^{-1}$) and $\boldsymbol{e_{p,90}}$ (in % of $max(x_{p,train})$), for each predictor $p$. These values were considered to represent a severe case for possible
model deficits due to lacking extrapolation capability. We created a respective test datasets for the sensitivity analysis from the 14-year overlap period (2007 – 2020): We selected the corresponding number of days $\boldsymbol{n_{ov}}$ with the highest values of the respective predictor in the, as

$$n_{ov} = \overline{n_{p,90}} \cdot 14 \, a \quad \text{(rounded to whole days)}, \tag{3}$$

and altered them by adding or subtracting the respective $e_{p,90}$. For example, $n_{p,90}$ of Q in Vent is 0.55 d a$^{-1}$, therefore
we changed $n_{ov} = 0.55 \cdot 14 \approx 8$ days by the $e_{p,90}$ of 9.6 m³ s$^{-1}$ (Table 2).

We used the resulting altered time series of the primary predictors to compute the corresponding ancillary predictors (that describe antecedent conditions, see section 2.1), ran the QRF model with them and compared mean annual SSC after the alterations to the original dataset. Thus, we performed six individual runs for the sensitivity



assessment at each gauge, two (one where the predictor was increased and one where it was reduced) for each of
the three primary predictors Q, P and T. We chose to compare mean annual SSC instead of annual SSY, as
discharge is needed to compute SSY so that the alterations in Q would have affected the estimated SSY twice.

*Table 2. Amount of reduction/increase in in the sensitivity models (average exceedance extent $\overline{e_p}$ in units of the corresponding predictor) and number of days with reduction/increase on average per year ($\overline{n_p}$) and in total in the 14-year period ($n_{ov}$).*

| | Q | | | P (summer) | | | T | | |
|---|---|---|---|---|---|---|---|---|---|
| | $e_{Q,90}$ [m³ s⁻¹] | $n_{Q,90}$ [d a⁻¹] | $n_{ov}$ [d] | $e_{P,90}$ [mm] | $n_{P,90}$ [d a⁻¹] | $n_{ov}$ [d] | $e_{T,90}$ [°C] | $n_{T,90}$ [d a⁻¹] | $n_{ov}$ [d] |
| **Vernagt** | 1.5 | 0.32 | 4 | 28.86 | 0.45 | 6 | 3.3 | 2.75 | 38 |
| **Vent** | 9.6 | 0.55 | 8 | 25.95 | 0.54 | 8 | 2.02 | 0.22 | 3 |

**Classification**

Third, we assessed whether the sensitivity analysis was informative for the different RCPs, time slices and
predictors, i.e. if the sensitivity analyses contained sufficiently extreme conditions to represent the projections. For
this, we determined the mean exceedance extent per predictor p, emission scenario rcp and time slice ts $\overline{e_{p,rcp,ts}}$
and the mean number of OOOR days per year $\overline{n_{p,rcp,ts}}$. We compared these to the $e_{p,90}$ and $n_{p,90}$, and marked
the respective predictor-time slice-RCP combination yellow, if ≥ 1/3 of projections had $\overline{n_{p,rcp,ts}} > n_{p,90}$ or
$\overline{e_{p,rcp,ts}} > e_{p,90}$, and red if this applied to ≥ 2/3 of the projections (see Figure 5 and Table 5).

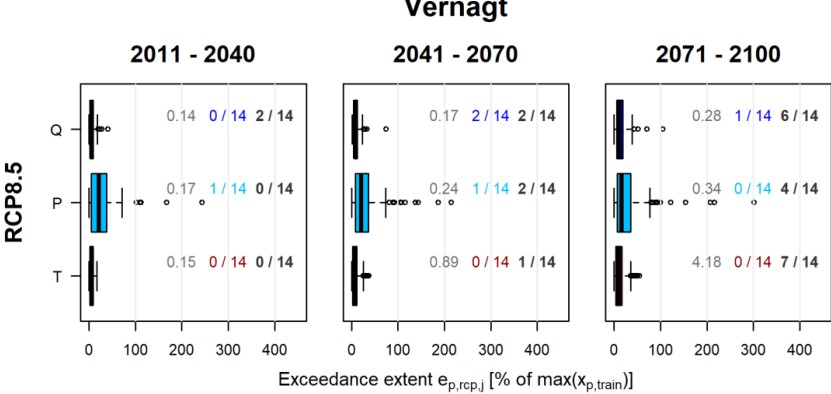

*Figure 5 Example of the classification based on the OOOR analysis. The boxplots show the distribution of exceedance extents $e_{p,rcp,j}$ per RCP and predictor on all days j within the respective time slice ts. Grey numbers denote the average $\overline{n_{p,rcp,ts}}$ of all projections within the respective RCP, time slice and predictor. Colored numbers indicate the number of projections with mean exceedance extent $\overline{e_{p,rcp,ts}} > e_{p,90}$, i.e. projection more extreme than the sensitivity analysis. Black numbers indicate the number or projections with the mean number of OOOR days per year $\overline{n_{p,rcp,ts}} > n_{p,90}$, , i.e. projection more extreme than the sensitivity analysis.*

**2.5    Analysis of model results**

We analyzed the model results, i.e. estimated annual yields in the application period for trends as well as shifts in
seasonality. To assess trends, we used two methods implemented in the R package *FUME* (Santander Meteorology
Group, 2012): the Mann-Kendall test, which is a non-parametric tool to detect linear trends (specifically, we used
a version that was modified to detect trends in serially correlated time series (Madsen et al., 2014; Yue et al.,



2012)) and Sen's slope estimator (Sen, 1968) to assess trend magnitude. Further, we compared the estimated yields in three time slices (near future: 2011 – 2040, intermediate future: 2041 – 2070, and far future: 2071 – 2100; see also Figure 2), comparable e.g. to Jacob et al. (2014) and Hanzer et al. (2018). To assess changes or shifts in the seasonality of sediment export, we compared the mean monthly yields of the observations and the projections.

To assess whether the detected trends are sensitive to the potential underestimation of yields on OOOR precipitation days, we multiplied the daily yields estimated by our QRF model on these days by a factor of 5, i.e. assuming a very severe underestimation in the original estimates (Figure 4). We chose this factor, as it is close to the most severe exceedance extents in precipitation at both gauges, which are 456 % at gauge Vernagt and 442 % at gauge Vent (see also section 3.3). We then compared the trends in annual SSY of the altered time series to the trends in the original QRF estimates. All analyses were conducted with the statistical software R (R Core Team, 2018).

## 3    Results

### 3.1    Verification of bias-corrected discharge for the present climate (2007 – 2020)

The bias-corrected discharge data yield more adequate representations of measured monthly discharge amounts and their seasonal distribution (Figure 6), as well as mean annual discharge volumes (Table 3). At gauge Vernagt for example, maximum mean monthly Q in the observations and the bias-corrected data is in August, whereas the original AMUNDSEN simulations suggested a maximum in July. Nevertheless, some underestimation of August discharge remains at gaute Vernagt. At gauge Vent, the original AMUNDSEN simulations substantially overestimated discharge amounts in April to July, i.e. the snowmelt period, which was successfully corrected by the bias-correction.

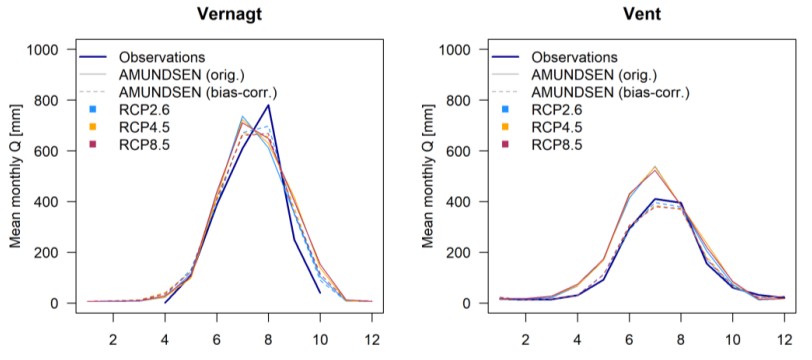

*Figure 6 Comparison of mean monthly discharge [mm] at gauge Vernagt (left) and Vent (right) derived from measurements, unaltered multi-model means of the original AMUNDSEN output and multi-model means of the bias-corrected AMUNDSEN output in the overlap period (2007-2020).*

*Table 3 Comparison of mean annual discharge volumes based on the original AMUNDSEN output, observations and bias-corrected AMUNDSEN estimates in the overlap period (2007 - 2020).*

| Mean annual Q (2007 – 2020) [mm] | Vernagt | Vent |
|---|---|---|
| **AMUNDSEN (orig.)** | 2530 | 1990 |
| **Observations** | 2310 | 1537 |
| **AMUNDSEN (corr.)** | 2400 | 1555 |





## 3.2 Verification of modeled SSY for the present climate (2007 – 2020)

We find good agreement between observed and modeled annual SSY at both gauges (Figure 7), and the Kolmogorov-Smirnov test does not yield significant differences between the observations and model results in mean annual sediment yields. Nevertheless, years with extremer annual yields (both lower and higher) occur in the model results, especially under RCP4.5 and RCP8.5 (e.g. for Vent, max. 3250 t a$^{-1}$ in RCP4.5 vs. 2120 t a$^{-1}$ in the observations), likely due to the higher sample size in the projections (42 or 196 years in the projections compared to 14 years in the observations, see also description of Figure 7).

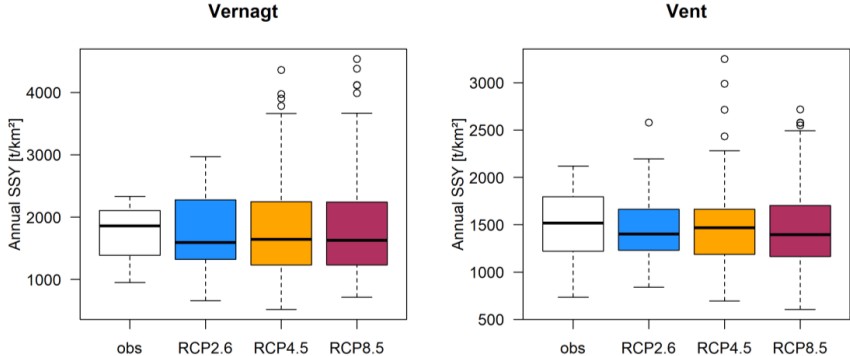

*Figure 7 Comparison of annual specific SSY in the overlap period (2007 - 2020) derived from measurements ("obs", n = 14 years) and QRF modelling results per RCP (n = 42 for RCP2.6 and n = 196 for RCP4.5 and RCP8.5, resp.) at gauges Vernagt (left) and Vent (right).*

Similarly, the seasonality of sediment export is well represented in overlap period of the projections (Figure 8), and the Kolmogorov-Smirnov test does not yield significant differences to the seasonal distribution of the measurements. Monthly SSY tend to be slightly lower in the projections in August at gauge Vernagt and in July and August at gauge Vent. Similar patterns had already become apparent in the comparison of mean monthly discharges at gauge Vernagt (Figure 6).

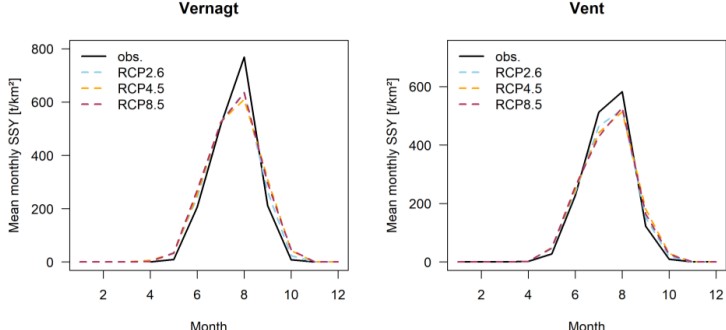

*Figure 8 Comparison observations (see also Figure 2) to QRF model forced by climate projections (multi-model means per emission scenario) during the overlap period (2007-2020).*



### 3.3 Assessment of limits of applicability

**Out-of-observation-range days**

Generally, we find more frequent OOOR days and higher exceedance extents in later time slices and in the higher emission scenarios (Figure A 1). At both gauges, OOOR days in Q are relatively rare, but in the higher emission scenarios and later time slices, OOOR days become more frequent and individual exceedance extents of more than 100 % occur (Figure A 1 and Table 4). Exceedances in temperature are more frequent at gauge Vernagt, especially under RCP8.5 and after 2040. At gauge Vent, there are only few OOOR days in T, except under RCP8.5 after 2070.

OOOR data points in summer precipitation are rather rare at both gauges. However, precipitation shows very high exceedances extents of up to ca. 450 % (RCP4.5 after 2070 and RCP8.5 before 2040 at gauge Vent; RCP4.5 after 2070 at gauge Vernagt, Figure A 1 and Table 4). This corresponds to daily precipitation sums of approx. 280 and 240 mm/day at gauge Vent and Vernagt, respectively, and is equivalent to over a third of the current mean annual precipitation at gauge Vent (687 mm (Hydrographic yearbook of Austria, 2016)). Yet even without the most extreme cases, exceedance extents in precipitation can be quite severe, which corresponds to very heavy precipitation events.

*Table 4 Mean and maximum exceedance extents $e_p$ of the three primary predictors discharge (Q), precipitation (P) and air temperature (T) across all emission scenarios and time slices, in percent of the maximum during the training period $max(x_{p,train})$ and original units.*

| $e_p$ | | Q [%] | Q [m³ s⁻¹] | P [%] | P [mm] | T [%] | T [°C] |
|---|---|---|---|---|---|---|---|
| Vernagt | Mean | 10.1 | 0.7 | 29.3 | 12.8 | 10 | 1.4 |
| | Max | 105 | 6.9 | 456 | 199.8 | 54.4 | 7.9 |
| Vent | Mean | 11.8 | 4.4 | 24.8 | 13.1 | 4.2 | 0.9 |
| | Max | 123 | 45.4 | 442 | 233 | 16.3 | 3.6 |

**Sensitivity analysis**

Given the alterations corresponding to the respective $n_{p,90}$ and $e_{p,90}$, the most sensitive predictor is P at gauge Vernagt and Q at gauge Vent (Figure 9). Yet although the reductions on the days with the highest P in the overlap period were quite substantial (Table 2), the maximum effect on the mean annual SSC is ≤ 3% at both gauges. Temperature is the second most sensitive parameter at gauge Vernagt, while the alterations at gauge Vent had little effect on mean annual SSC. At gauge Vent, P is the second most sensitive parameter, but with a maximum effect of < 2% on mean annual SSC.

The results of the sensitivity analysis also give indication of the behavior of the QRF model in response to OOOR data points: as expected, we generally observe a decrease in mean annual SSC if we decrease the predictors (Q, P and T), and vice versa. However, for most predictors, the decrease is more pronounced than the increase (although the same days were altered by the same extent). We presume that this is due to the described incapability of QRF to extrapolate. Thus, we can expect to underestimate the additional effect, e.g. of precipitation exceeding *max(P_{train})*.





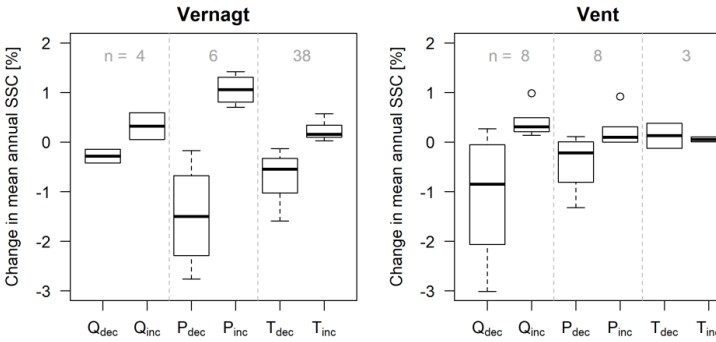


*Figure 9 Results of the sensitivity analysis for gauges Vernagt (left) and Vent (right) with respect to mean annual SSC of the years with altered days. Subscripts signify increase (inc) and decrease (dec) in the predictors by the respective average exceedance extent $e_{p,90}$ and frequency $n_{p,90}$ as identified based on the exceedance analysis. Grey numbers represent the number of altered days.*

## Classification


Table 5 shows, that until 2070, all predictors and RCPs fall within the conditions covered by the sensitivity analysis (with the exception of Q after 2040 at gauge Vent under RCP8.5). This implies that the results of the sensitivity analysis are informative in these cases, and that we expect similar or smaller effects of OOOR days on mean annual SSC or SSY in the application period. After 2070, exceptions occur at both gauges in two out of three RCPs and

several predictors, which implies that the uncertainty is higher than in the results of the sensitivity analysis.

*Table 5 Results of the classification per emission scenario, predictor and time slice. The color of the field denotes more than 1/3 (yellow) and 2/3 (red, does not occur) of the projections with $\overline{e_{p,rcp,ts}} > e_{p,90}$ (assumed in sensitivity); N denotes more than 1/3 (yellow) and 2/3 (red, does not occur) of the projections with $\overline{n_{p,rcp,ts}} > n_{p,90}$.*

| | | Vernagt | | | Vent | | |
|---|---|---|---|---|---|---|---|
| | | 2011 – 2040 | 2041 – 2070 | 2071 – 2100 | 2011 – 2040 | 2041 – 2070 | 2071 – 2100 |
| **RCP2.6** | **Q** | | | | | | N |
| | **P** | | | N | | | |
| | **T** | | | | | | |
| **RCP4.5** | **Q** | | | | | | |
| | **P** | | | | | | |
| | **T** | | | | | | |
| **RCP8.5** | **Q** | | | N | | N | N |
| | **P** | | | N | | | |
| | **T** | | | N | | | N |

The OOOR analysis showed that very high exceedance extents occur in precipitation and that precipitation is a sensitive parameter at both gauges (although the effect on mean annual SSC was small). Additionally, we find that heavy summer precipitation becomes more intense (and only slightly more frequent) (Figure A 2 in appendix): the 99.95th percentile of the summer precipitation projections increases over time, which suggests that precipitation events become more intense. At the same time, the number of precipitation events that exceed the 99.95th percentile

determined from the precipitation observations in the overlap period (2007 – 2020) hardly increases on average,



which suggest that precipitation events of a certain strength do not become (much) more frequent. We also find an increase in daily SSY associated with heavy precipitation events (Figure A 2 in appendix).

Thus, we additionally assessed whether the trends in annual yields were sensitive to changes in yields on days with OOOR precipitation (sections 2.5 and 3.4.1), as extreme precipitation can be very important for sediment dynamics

(e.g. by triggering mass movements).

### 3.4    Projections of future sediment export: changes in annual yields, timing of peak sediment and changes in seasonality

#### 3.4.1    Changes in annual yields and timing of peak sediment

The resulting projections suggests an overall decrease in mean annual SSY for both gauges and each of the three

emission scenarios, which is more pronounced at gauge Vernagt (Figure 10 and Table 6). Accordingly, we consistently find significant negative trends in the projections (2007 – 2100) in mean annual SSY (Table 6). The differences between the RCPs are small, and smaller than the spread within individual RCPs (Figure 10). Accordingly, trends of mean annual SSY are only slightly more negative in the high-emission scenarios. With respect to the 99th percentile of annual SSY estimates, trends are less strong than for mean SSY estimates at gauge

Vent, while at gauge Vernagt, the trends in the 99th percentile are even stronger than for mean annual SSY estimates.

Negative trends were detected for all individual projections as well: at gauge Vent, 26 (out of 31) are significant ($\alpha = 0.05$, Sen's slope ranging from -10.8 to -3.8 t km$^{-2}$ a$^{-2}$), and at gauge Vernagt, 30 of 31 are significant a ($\alpha = 0.05$, Sen's slope ranging from -15.2 to -6.1 t km$^{-2}$ a$^{-2}$).

*Table 6 Trends in mean and 99th percentile of annual specific SSY projections (2007 – 2100) given as Sen's Slope [t km$^{-2}$ a$^{-2}$] for the original estimates and the altered estimates (5-fold increased SSY on days with OOOR precipitation). Significance levels: \* = 0.05, \*\* = 0.01, \*\*\* = 0.001.*

| Sen's Slope of mean annual SSY [t km$^{-2}$ a$^{-2}$] | | RCP2.6 | | RCP4.5 | | RCP8.5 | |
|---|---|---|---|---|---|---|---|
| | | Original | Altered | Original | Altered | Original | Altered |
| **Vernagt** | Mean | -10.6*** | -10.7*** | -11.6*** | -11.5*** | -12.4*** | -12.1*** |
| | 99 percentile | -12.3*** | -12.3*** | -22.6*** | -22.3*** | -22.3*** | -21.3*** |
| **Vent** | Mean | -4.85 ** | -4.97** | -5.0*** | -4.67*** | -6.41*** | -5.6*** |
| | 99 percentile | -4.5** | -5.1*** | 0.5 | 2.3 | -3.1* | 0.1** |

The trend in the altered time series (with 5-fold increased daily yields on days with OOOR precipitation; see

section 2.5) hardly differs from the trend in the original time series (Table 6). Specifically, at gauge Vernagt, trend characteristics are basically unchanged. The only trend reversal occurs in the 99th percentile at gauge Vent under RCP8.5, where the trend is slightly positive (and significant) instead of negative. We conclude that the overall trend characteristics remain very robust, even if we assume as severe underestimation of the model on days with OOOR values in the predictors. Thus, the overall future sediment budget seems to be governed by their mean

behavior rather than solitary extreme events.



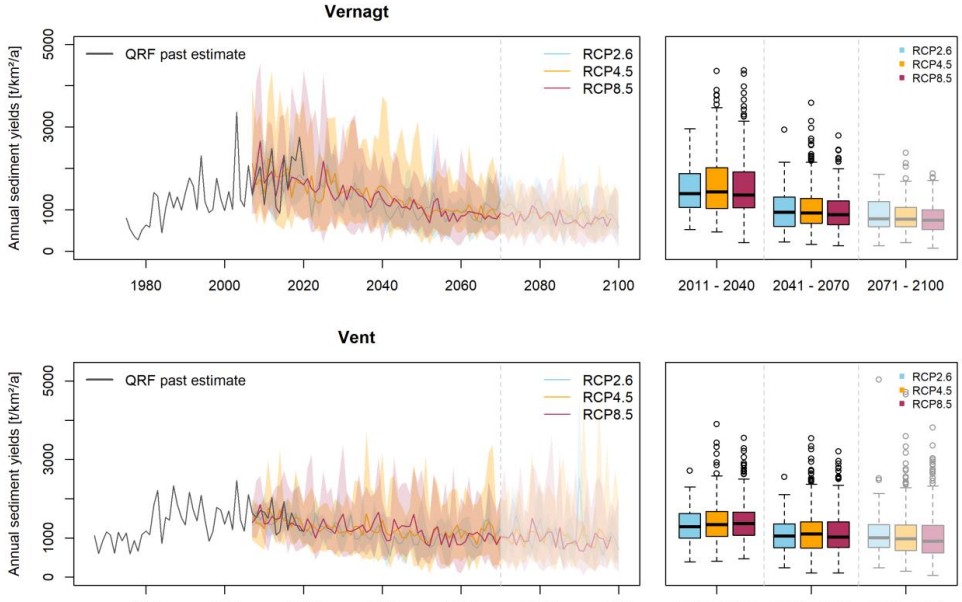

*Figure 10 Left: Mean annual suspended sediment yields per RCP, with minima and maxima of the individual projections indicated by the colored envelopes. Right: Annual SSY of all respective years and projections within the three time slices.*

The synopsis with estimates of annual SSY for the past decades shows that we find increases in annual yields at both gauges up until sometime between 2000 and 2020, and decreases afterwards, which is much more distinct at gauge Vernagt (Figure 10). This suggests, that 'peak sediment' has already been reached or is underway at both gauges and occurs simultaneously with 'peak water'.

### 3.4.2 Changes in seasonality

Mean monthly SSY is projected to decrease substantially during the glacier melt period in August in all RCPs and at both gauges (Figure 11). This decrease amounts to approx. ⅓ to ½ at gauge Vernagt and approx. ⅓ at gauge Vent. As a result, the highest mean monthly SSY shifts from August to July, or even to June under RCP8.5 after 2070 at both gauges. Additionally, the spring onset of sediment export is projected to occur earlier in the year in the high emission scenarios. This represents a decrease in importance of glacier melt for sediment export. After 2070, only relatively minor further changes are projected under RCP2.6 and RCP4.5, whereas RCP8.5 experiences further decreases in mean monthly SSY throughout the year.

At gauge Vent, a slight increase in mean July SSY is projected after 2070 under RCP2.6. This is likely related to an increase in discharge, since this increase is not visible in mean monthly concentrations (not shown). It also has to be considered that only three projections are averaged for RCP2.6 (as compared to 14 in the other RCPs), which makes it less robust with respect to outliers.

A comparison to the seasonal distributions determined from the altered time series (5-fold increased SSY on days with OOOR precipitation), showed only very slight differences, which indicates that the seasonal distribution is also insensitive to underestimations of SSY on days with heavy precipitation.



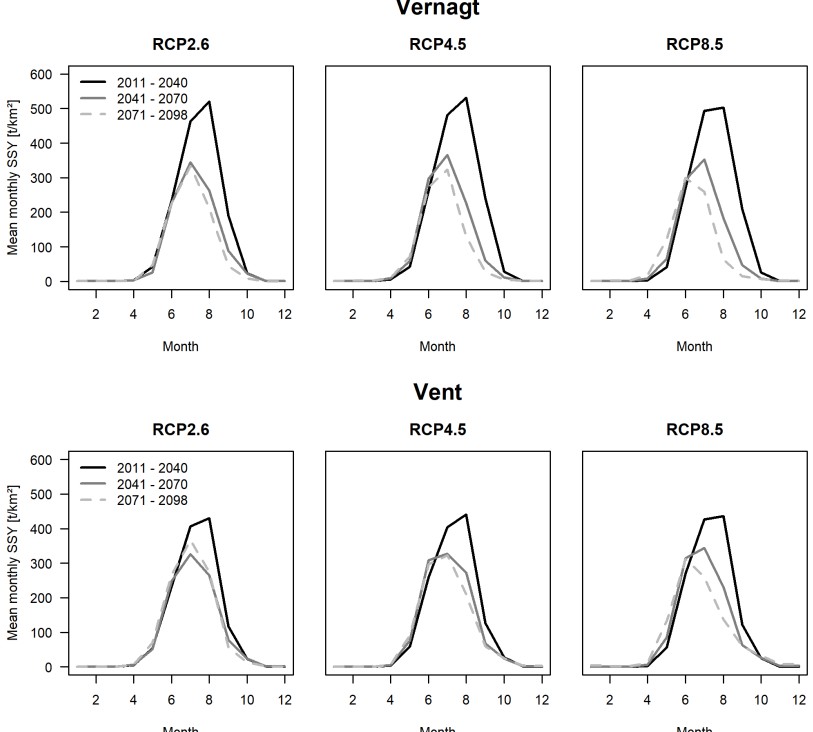

*Figure 11 Seasonality of mean monthly SSY in three time slices and emission scenarios.*

## 4    Discussion

Testing new methods to estimate future suspended sediment export from glacierized high-alpine areas can provide important information, e.g. to flood hazard, sediment or water quality management, since estimating such changes had so far been limited to relatively rough approximations. This study represents the first attempt to our knowledge to derive SSY projections using a machine-learning approach and investigate them in synopsis with reconstructed past SSY.

**Projected changes in sediment export and location of peak sediment**

The presented SSY projections in the Ötzal, Austria, suggest an overall decrease in annual SSY. This is consistent across emission scenarios as well as all individual projections (i.e. based on the 31 different RCP-GCM-RCM-chains). The decrease is much more distinct at gauge Vernagt, where a trend analysis in the previous study (Schmidt et al., 2023) showed significant positive trends in the period until 2020. At gauge Vent, significant positive trends were detected if all data points since the 1970s are considered (ibid.). However, if only the years after the distinct increase around 1980 were considered, the trend was slightly negative (ibid.). This suggests that 'peak sediment' has already been reached or is underway at both gauges and occurs simultaneously with 'peak water'.





These findings match expectations of conceptual models, that sediment yield from deglaciating basins will initially increase (due to increases in glacial erosion, sediment supply accessibility, transport capacity and occurrences of extreme floods) and subsequently decrease, as glacier masses decline, meltwater volumes and freeze-thaw weathering decrease, and vegetation colonizes (Antoniazza and Lane, 2021; Zhang et al., 2022). It is expected that peak sediment may lag behind peak meltwater, with a lag that can be up to decades or centuries (Delaney and

Adhikari, 2020). This lag is hypothesized to be scale-dependent, i.e. will be shorter for areas closer to the glacierized regions, and to depend on the changes in erosive precipitation: if erosive precipitation decreases, peak sediment occurs simultaneously with peak water, while increasing or stable erosive rainfall scenarios are associated with a lag (Zhang et al., 2022). Indeed, for the study area of this study, a decrease in summer precipitation sums (i.e. June to August, which is the time of minimum snow-cover and thus maximum erodibility)

is projected (Hanzer et al., 2018), and our estimates suggest that 'peak sediment' coincides with 'peak meltwater'.

Sediment export projections differed only slightly (if at all) between emission scenarios, i.e. the spread between projections within one emission scenario is much larger than differences between ensemble means of the three RCPs. It should be noted, that comparisons to RCP2.6 need to be treated with care, as it comprises less GCM-RCM combinations (only 3 as compared to 14 in the higher emission RCPs). Nevertheless, the absence of major

differences between RCPs is in accordance with findings by Gobiet & Kotlarski (2020), that "until the middle of the 21st century [...] it is projected that climate change in the Alpine area will only slightly depend on the specific emission scenario." Accordingly, Hanzer et al. (2018) projected glacier volumes to decline by 60-65 % until 2050 "largely independent of the emission scenario".

### Changes in seasonality and response to heavy precipitation events

Despite the overall decrease in SSY, our results suggest that high annual SSY are possible, especially at gauge Vent and towards the end of the century, and that yields on days with heavy precipitation may increase at both gauges – in absolute terms and in relation to the annual export. This is reasonable, given that increases in heavy precipitation intensity and/or frequency in the European Alps have been detected in measurement data from the past (e.g. (Hiebl and Frei, 2018; Scherrer et al., 2016; Gobiet and Kotlarski, 2020)), as well as future projections

(Gobiet and Kotlarski, 2020; Jacob et al., 2014; Kotlarski et al., 2023) – despite the overall decrease in summer precipitation mentioned above. As a result, we can expect an increase in sediment-related harmful events triggered by heavy precipitation, such as flash floods and gravitational mass movements (i.e. debris flows, landslides) (Huggel et al., 2012; Savi et al., 2020; Gobiet and Kotlarski, 2020). Similar expectations, i.e. increasing high-magnitude transport events in the context of an overall decrease, have e.g. been expressed with respect to bedload

in South Tyrol (Coviello et al., 2022). Such a development would have important implications e.g. for flood hazard management, as the flood risk can increase if cross-sections are reduced after sedimentation and potential backwater effects need to be considered (Nones, 2019), and where much of the damage is associated with transported solids rather than the water itself (Badoux et al., 2014).

Our findings suggest a shift in sediment export seasonality, since the highest mean monthly SSY shifts from

August to July (or even June), due to substantial reductions in sediment export in July, August and September at both gauges. This is linked to the projected distinct reductions in glacier melt (Hanzer et al., 2018) and appears reasonable given that glacier melt has so far been the dominant transport medium of suspended sediments at these gauges (Schmidt et al., 2022). These results are not sensitive to potential underestimations of SSY on days with



very heavy precipitation. Such shifts in seasonality and the concomitant overall reduction in fluvial sediment
transport will likely have severe effects on biodiversity, i.e. flora and fauna of glacier-fed streams (Milner et al.,
2009, 2017; Gabbud and Lane, 2016; Huss et al., 2017).

**Limitations**

As a potential limitation to the presented Quantile Regression Forest approach, out-of-observation-range data
points in the predictors can lead to underestimates in SSY on the affected days. Yet, the analysis of such incidents
in synopsis with the results of the sensitivity analysis showed that before 2070, the effect on annual yield estimates
is $\leq 3\%$. This is very small given the overall high variability in SSC (Vercruysse et al., 2017; Delaney et al., 2018;
Schmidt et al., 2022). On a similar note, even assuming rather generous increases of yields on days with OOOR
precipitation altered the trends only marginally, which shows that underestimations on individual days with OOOR
precipitation has little effect on long-term annual averages. However, we have less confidence in the model results
after 2070 for two reasons. First, more frequent and severe OOOR incidents occur during this time, especially in
the high-emission scenarios, and fewer projections are covered by the assumptions of the sensitivity analysis. We
can therefore expect a higher uncertainty in the model results. Specifically, the effect of underestimation for single
large events will aggravate. Second, more than a few glacier simulations suggest that glaciers could have
disappeared almost entirely by 2070 (Hanzer et al., 2018), which implies a major shift in the hydro-
sedimentological functioning of these catchments. While our QRF models were able to model threshold effects
better than sediment rating curves (Schmidt et al., 2023) (likely because they are not bound to linear or monotonous
relationships), this is only true for effects that are represented in the training data. Thus, the results for the period
after 2070 need to be treated with caution. We have indicated this in the presentation of our results by using
transparency or dashed instead of solid lines.

As a more general limitation, there are several other factors with the potential to substantially alter and influence
sediment dynamics in the study area, which we cannot consider in our models. This concerns geomorphological
changes, such as increased paraglacial erosion: debuttressed slopes may trigger landslides and rockfalls, and
indeed, increased debris flow and rockfall activity have been shown in response to warming in other areas, likely
associated with intensified alpine permafrost thaw (Savi et al., 2020; Hartmeyer et al., 2020; Huggel et al., 2012).
Additionally, sediment availability and accessibility increase as erodible landscapes expand (Li et al., 2021a), and
subglacial sediment availability might also increase (more subglacial sediment can be accessed by meltwaters as
the equilibrium line altitude retreats upslope) until the glacier size becomes smaller than a critical size (Delaney
and Adhikari, 2020; Zhang et al., 2022). Although these processes are likely already partially reflected in the
observations used for the model training, their intensity may still be too low to be sufficiently learned by the model.
Thus, future intensification of these processes could lead to higher sediment export rates than our estimates
suggest, and might thereby affect the estimated location of 'peak sediment'. Notwithstanding, there are also several
factors that could lead to decreases in sediment export, such as decreases in connectivity (such as the formation of
supra-, sub- or proglacial lakes or outwash fans which act as sediment traps) or decreasing glacial erosion as
glaciers recede (Zhang et al., 2022). Additionally, freeze-thaw weathering may decrease (Hirschberg et al., 2021)
and it is not clear how quickly the deglaciating landscapes will stabilize, e.g. through eluviation of fine materials
and fluvial sorting of sediment, which progressively increases the resistance to entrainment and transport
(Ballantyne, 2002; Lane et al., 2017), or vegetation colonization (Haselberger et al., 2021; Altmann et al., 2023;



Musso et al., 2020; Eichel et al., 2018). Many of these processes are ultimately governed by temperature and/or precipitation, and we have chosen the predictors to act as proxies (e.g. antecedent moisture and temperature conditions could be crucial for mass movements). While this is out of scope of the presented study, we encourage future studies to work towards including more advanced proxies for geomorphological changes.

**Uncertainties**

The presented results are associated with uncertainties, which are a combination of uncertainties inherited from the underlying climatological and hydrological projections and uncertainties inherent in the QRF approach. Climate model uncertainty represents a combination of uncertainties in assumptions of future anthropogenic greenhouse gas emission, GCM uncertainty (different GCMs produce different responses to the same radiative forcing) and RCM uncertainty (different RCMs forced by same GCM produce different regional responses) (Evin et al., 2021; Gobiet et al., 2014). It has been found that EURO-CORDEX simulations may be biased towards "too cold, too wet, too windy", but that these uncertainties are mostly within the observational uncertainties, and that simulations "reproduce fairly well the recent past climate despite some biases" (Vautard et al., 2021). To address this, it was recommended to carry out bias-correction, which has been performed by means of quantile mapping for the precipitation and temperature projections (Hanzer et al., 2018). The hydrological model results are also associated with uncertainties, such as the tendency to overestimate spring runoff, winter snow accumulation and glacier mass balances. We have addressed this through bias-correcting the discharge projections, which resulted in a more adequate representation of discharge seasonality and volumes. Certainly, bias-correction methods such as quantile mapping in turn introduce uncertainties, e.g. by assuming that the biases are stationary, i.e. do not change over time (Gudmundsson et al., 2012). Hydrological simulations that do not show the necessity for this correction could eliminate this issue. Uncertainties in the QRF approach have been addressed in a previous study, and include the tendency to underestimate rare, high-magnitude daily SSY (albeit with small effects on the respective annual yields), the underestimates on days with OOOR values (which had small effects until 2070, as discussed in detail above) and the choice of temporal resolution (i.e. daily compared to hourly resolution involves some loss of information, e.g. on precipitation intensities, but the effect was also found to be small) (Schmidt et al., 2023). Since QRF is a data-driven approach, the quality of the estimates hinges on the underlying training data set as well as the choice of predictors, i.e. a large and varied enough dataset in combination with predictors that meaningfully represent the most important processes improve the quality of the estimates (ibid.). Thus, future studies are recommended to explicitly sample extreme events, and/or verify the representativity of the training dataset.

## 5    Conclusion

We found decreasing trends in annual SSY at both gauges regardless of the emission scenario, which suggests that peak sediment was already reached between 2000 and 2020. These findings persist even if yields on days with projected heavy precipitation are inflated by a factor of five. Despite the projected overall decrease, high(er) annual yields are possible, likely in response to heavy summer precipitation. This discrepancy has important implications for sediment management, but also e.g. of flood management.



To our knowledge, this study represents the first attempt to combine machine learning for suspended sediment modeling with climate and hydrological projections, in order to derive projections of sediment export in high-alpine areas. It demonstrated that Quantile Regression Forest can be a valuable tool for this application. We addressed known issues of QRF, i.e. underestimations on days where predictors in the application period exceed the range represented in the training data. The influence on the results showed to be negligible until 2070. We conclude that the presented results are much more uncertain after 2070, partly because of more frequent and severe out-of-observation-range data points, but mainly since a major shift in the functioning of the hydro-sedimentological system can be expected as deglaciation is quasi completed.

However, while the chosen predictors represent proxies for crucial processes controlling sediment transport in these high-alpine environments, several potentially crucial geomorphological factors, that could increase or decrease sediment export (and thereby change the projected trends and location of peak sediment) could not be taken into account. These include increases in rockfalls and landsliding, changes in connectivity or vegetation colonization. Future studies are encouraged include these factors more explicitly.

**Code and data availability**

The model code, input data and results are published under DOI 10.23728/b2share.5f706863fc5041c49cb8d1a8cd55f613.

**Author contributions**

LKS developed the general idea and conceptualized the study with TF and PG, with mentoring and reviewing of AB. LKS requested the necessary input data, PG adapted and extended the model code under supervision of TF and LKS, and PG and LKS performed the model runs. LKS developed and conducted the analysis of model limitations and conducted the statistical analyses with support and supervision by TF. LKS prepared the original draft of the manuscript including all figures, and all authors contributed to writing this paper.

**Competing interests**

The contact author has declared that none of the authors has any competing interests.

**Acknowledgements**

This research was funded within the DFG Research Training Group "Natural Hazards and Risks in a Changing World" (grant nos. NatRiskChange GRK 2043/1 and GRK 2043/2) and by a fieldwork fellowship of the German Hydrological Society (DHG).

The authors would like to thank Florian Hanzer and Ulrich Strasser for providing the projections required as input data for this study. We thank the Hydrographic Service of Tyrol, Austria, and the Bavarian Academy of Sciences and Humanities for the provision of data as well as logistical support. We thank Marvin Teschner for his support in data analyses.



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



**Appendix**

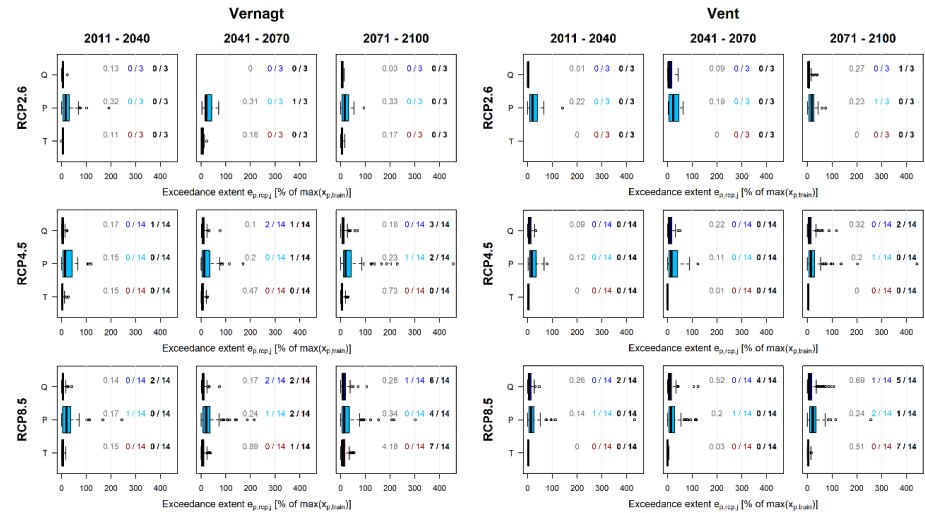

960

*Figure A 1 Results of the classification, based on the sensitivity and OOOR analyses. The boxplots show the distribution of exceedance extents for each RCP and predictor on all days j within the respective time slice ts. Grey numbers denote the average $\overline{n_{p,rcp,ts}}$ of all projections within the respective RCP, time slice and predictor. Colored numbers indicate the number of projections with mean exceedance extent $\overline{e_{p,ts}} > e_{p,90}$ (as used for the sensitivity analysis). Black numbers indicate the*

965 *number or projections with the mean number of OOOR days per year $\overline{n_{p,ts}} > n_{p,90}$ (as used for the sensitivity analysis).*

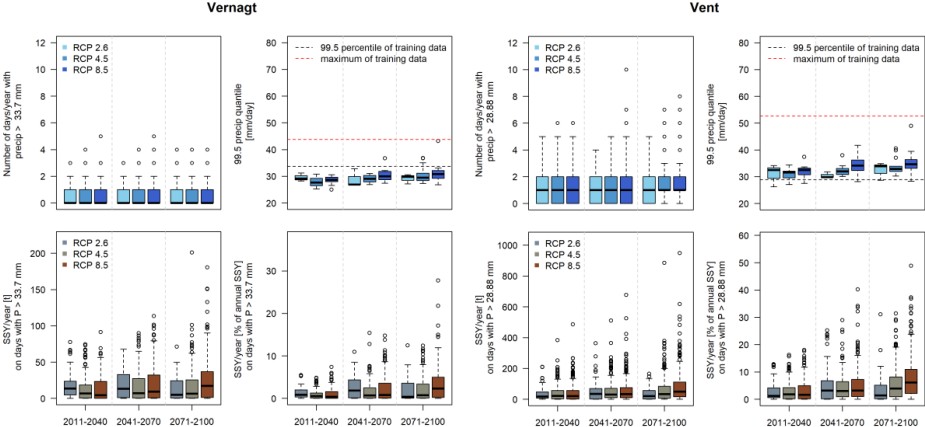

*Figure A 2 Analysis of summer precipitation projections (top) and SSY projections (bottom) at gauges Vernagt and Vent. Top left of each panel: frequency of days with heavy summer precipitation (> 99.5 percentile of the training data, i.e. and 33.7*

970 *mm/d (Vernagt) and 28.88 mm/d (Vent)). Top right: intensity of heavy summer precipitation events over time, expressed as the 99.5 percentile. Bottom left: Sediment export on days with precipitation > 99.5 percentile of the training data. Bottom right: Sediment export on days with precipitation > 99.5 percentile of the training data relative to the respective annual yields*