# Peer review of "Projecting sediment export from two highly glacierized alpine catchments under climate change: Exploring non-parametric regression as an analysis tool"

_EGUsphere, 2023_

## Author Comment (AC2)

Figures with references to answers to anonymous referee #2

[Figure]

Figure 1 Changes in Q seasonality in AMUNDSEN results without bias-correction (from figure 15 in Hanzer et al., 2018)

[Figure]

Figure 2 Changes in seasonality in bias-corrected AMUNDSEN discharge projections, i.e. after quantile mapping.

[Figure]

*This will be the new figure 11, showing the estimated timings of peak water and peak sediment. Black lines indicate past mean annual Q from measurements and mean annual SSY estimates of QRF model; colored lines correspond to different RCPs (compare to Figure 10). Underlying data have been smoothed using a 15-year moving average.*

---

## Author Response (AR1)

*Dear Manuela Brunner, dear Mustafa Al-Mukhtar, dear anonymous referees,*

*We would like to thank you for your detailed comments, questions, suggestions, and constructive feedback. Below, we provide our responses as direct answers to each comment and point out our suggested changes to the manuscript. (Please note that the line numbers refer to the pdf of the revised manuscript unless otherwise mentioned.)*

*We hope that these suggestions will be to your satisfaction.*
*Best,*

*Lena Katharina Schmidt on behalf of all authors*

**Mustafa Al-Mukhtar, 21 Jul 2023**

I have read the manuscript entitled "Projecting sediment export from two highly glacierized alpine catchments under climate change: Exploring non-parametric regression as an analysis tool" submitted to HESS for possible publishing. The authors' presents the quantile regressing forest QRF to model the future (up to 2100) suspended sediment yield from two Alpine catchments. The authors used to that ends climatological data represented by the air temperature and precipitation from the European part of the COordinated Regional Downscaling Experiment (EURO-CORDEX) and for the discharge projections they used the physically-based hydroclimatological and snow model AMUNDSEN. They quantified the uncertainty inherited from the model structure and data by the OOOR method. The paper is well written and structured. It was hardly to detecting any language error or typo. In my perspective the paper is worth publishing in HESS.

***Answer:** Thank you for this reassuring review.*

**Anonymous Referee #2, 27 Jul 2023**

**Projection of sediment yield has been a long-lasting challenge. This study gives a meaningful attempt to project sediment export from two highly glacierized alpine watersheds by applying machine-learning approaches. Although the projection results of sediment yield highly rely on the accuracy of input data, e.g., climate scenario data and hydrological simulations and bias-corrected methods, the method framework in this study provide a useful reference and has great potential to help predict how future climate will impact the future sediment transport. Although this manuscript is well written and the limitations and uncertainties have been discussed, I still have some minor to moderate comments that would help to improve the quality of this manuscript.**

**Major concern:**

Please justify that the current bias-corrected method is reasonable in correcting hydrological data. I find the current bias-corrected method used in section 3.1 and figure 6 may erase the future changes in season hydrological patterns when it is applied to discharge projection data. For example, it will artificially keep the future seasonal pattern of discharge the same as that

between 2007–2020 and bury the information on regime shifts in hydrological processes. Especially in glacierized watersheds, the month of the maximum runoff may be changed from one to another with the release of glacier melt water and glacier retreat, thus shifting the seasonal pattern of discharge. However, this information seems to be distorted by the current bias-corrected method.

*__Answer:__ Thank you for your concern. As the figures below show, the changes in Q seasonality are preserved well after quantile mapping: figure 1 shows the seasonality and its changes over time as published by Hanzer et al., i.e. on the Q projections before quantile mapping. Figure 2 shows the seasonality of the bias-corrected Q data. Absolute volumes have changed a little (which was the point of doing the quantile mapping), but the seasonal pattern is preserved well.*

[Figure]

*Figure 1 Changes in Q seasonality in AMUNDSEN results without bias-correction (from figure 15 in Hanzer et al., 2018)*

[Figure]

*Figure 2 Changes in seasonality in bias-corrected AMUNDSEN discharge projections, i.e. after quantile mapping.*

The sensitivity analysis shows that the most sensitive predictor is different in two watersheds. Does this mean that the major control of sediment transport processes is different in these two watersheds? Can you explain this?

***Answer:*** *Thank you for this important feedback. Unfortunately, the interpretation of variable importance (which is essentially what you are suggesting) is not straightforward, because the predictors are correlated (for example some information on temperature is present in discharge, as they are linked through melt processes). In contrast, this sensitivity analysis is meant to indicate how influential the OOOR days could be for our results (for example, 38 altered days in temperature have a smaller effect on yields than 6 altered days in precipitation at gauge Vernagt, and the magnitude of changes in yields is very small in general). Additionally, a direct comparison between the gauges is not possible, because the number of altered days and the amount by which those days were altered for the sensitivity analysis is determined from the number of OOOR days and the exceedance extents. That means, this differs between the gauges, and the respective response of the catchments is not comparable.*

*We have stated more clearly the purpose of the sensitivity in the description around line 328 to avoid confusion.*

Can the QRF model trained in this study capture the non-linear effect of climatic factors, like temperature? For example, the initial increase in temperature can lead to an increase in annual SSC, but after the tipping point, the continuous warming can be accompanied by a decrease in SSC due to the exhaustion of sediment supply and depletion of glacial melt water. From the sensitivity analysis, it seems that SSC can only increase with the temperature as shown in Figure 9.

***Answer:*** *Thank you. We do think that the model can capture this – since it is not a univariate relationship between temperature and SSC and also not bound to linearity. In the scenario you outlined above, not only an increase in T but also the decrease in Q will affect the SSC estimations, and the depletion of glacial melt water is modeled in the discharge projections, which are based on a glacio-hydrological model. Additionally, our results from the previous publication indicated that the model can deal well with non-linear behavior, such as periods containing threshold-like effects due to extreme events.*

*Nevertheless, as we stress in the discussion (and indicate through transparency or dashed lines), we suggest to interpret the results for the period after 2070 with caution, as glaciers are projected to have vanished (almost) completely in most projections by then, which could fundamentally alter the functioning of the catchments with respect to sediments in a way that may not be learnable from the training data.*

**Specific comments:**

Line 70: The timing of 'peak sediment' is presumed to depend i.a. on changes in erosive precipitation. What does the 'i.a.' here stand for?

***Answer:*** *This refers to figure 6 in Zhang et al., 2022, on the timing of peak sediment relative to peak water and the completion of deglaciation. They differentiate between three different scenarios for erosive precipitation (increasing, stable and decreasing), each of which shows a wide range of possible timings and relative magnitudes of peak sediment (anything between peak meltwater and the completion of deglaciation is possible, or even before peak meltwater). This uncertainty must be caused by other factors, hence the inter alia. We understand that it might cause confusion here though, and have changed the sentence to "The timing of 'peak sediment' relative to 'peak meltwater' and the completion of deglaciation is presumed to depend on changes in erosive precipitation."*

Please be consistent in using the abbreviations. i.e, or e.g.

**_Answer:_** _In our understanding, these mean different things? We use i.e. to express "that is to say" or "namely", and e.g. to abbreviate "for example"._

Line 380 and figure 7: Are those years with extremely high annual SSY linked to extreme hydrologic or climatic events? Are there any extremely high discharges or precipitation during these outlier years?

**_Answer:_** _Yes, the years with the highest SSY also are the years with the highest annual Q. There is not such a clear relationship to mean annual temperatures, mean summer temperature, mean July temperature or the maximum daily precipitation in a year. But of course a prolonged period with medium heavy precipitation or relatively warm temperatures could also be considered an event. Since figure 7 summarizes the data of 448 (simulated) years (14 years of actual measurements and 41 + 196 + 196 years of simulations from the 31 projections), we suggest that analyzing all of them for events is out of scope for this manuscript, which focusses on long-term changes rather than individual events._

Line 390: the authors may need to clarify that the peak sediment yields or high values are underestimated although the overall seasonal pattern is captured.

**_Answer:_** _Thank you for your concern. We believe that the sentence addresses this (L406: "Monthly SSY tend to be slightly lower in the projections in August at gauge Vernagt and in July and August at gauge Vent."). More generally, we agree that high values may be underestimated, (hence the OOOR and sensitivity analyses). This point is already addressed in the discussion (first sentence under "limitations")._

Figure 10: I would suggest including the time series of annual discharge to back your statement that 'peak sediment occurs simultaneously with peak water'.

**_Answer:_** _thank you for this suggestion. As combining everything into one plot turns out to be too messy and hardly interpretable, we added another figure showing peak sediment and peak water at both gauges (see below). For this, we applied a 15-year moving average to the annual data, as is customary to visualize peak water (this allows for a clearer picture despite interannual variations)._

[Figure]

*The new figure 11, showing the estimated timings of peak water and peak sediment. Black lines indicate past mean annual Q from measurements and mean annual SSY estimates of QRF model; colored lines correspond to different RCPs (compare to Figure 10). Underlying data have been smoothed using a 15-year moving average.*

Line 485: More explanation is needed in interpreting the changes in SSY seasonality. For example, how the shifts of the highest mean monthly SSY and earlier onset of sediment transport are linked to future climate change and glacier changes?

***Answer:*** *We think some of the interpretation you are looking for can be found in the discussion around line 585: "This is linked to the projected distinct reductions in glacier melt (Hanzer et al., 2018) and appears reasonable given that glacier melt has so far been the dominant transport medium of suspended sediments at these gauges (Schmidt et al., 2023)." We included: "The earlier onset of sediment transport in spring in the high emission scenarios is likely due to the earlier onset of snow and glacier melt in the discharge projections (see also fig. 15 in Hanzer et al., 2018)."*

**Anonymous Referee #3, 28 Jul 2023**

Schmidt et al. use Quantile Regression Forest (QRF) with climate and hydro projections to assess climate change impacts on suspended sediment yields until the end of the century in two glacierized basins. This is an important and understudied topic for which a novel modeling framework is presented. Therefore, I find this work to be relevant for publishing in HESS. In my opinion, the manuscript is generally in a good state and the authors are in a good position to address my major comments below and the more specific ones further down.

**Major comments**

With the sensitivity analysis the authors made good job of understanding how their model works and how P/Q/T influence the results. However, for the future changes it seems like there is little interpretation on how these results emerge. No specific reason for these changes is mentioned in the abstract or conclusions but would be important. Also, there is no figure

showing the how the most important inputs develop in future.

*Answer: Thank you for this important comment. The changes in the predictors have been analyzed at depth by Hanzer et al., 2018. Thus, we added a summary of the projected changes to the discussion (around line 530). Changes in heavy precipitation are addressed in figure A2 in the appendix, which is described in the results (line 460 et seqq.). We added changes in Q in the new figure 11 (see above request by reviewer #2) and included a short summary for the reasons of SSY changes in the abstract and conclusion.*

The major limitation is the lack of geomorphic processes in the projections, but this is rightfully acknowledged and discussed. An interesting discussion (around L520) on the timing of peak water/melt/sediment comes up, which I think has more potential. First, as I see it the finding of the coinciding timing of these is not backed up with any data in the ms, so adding this data to a figure would be good. Second, a physical explanation on why these timings coincide is missing, but would be important especially because you use data-driven methods and not physically-based models.

*Answer: Thank you, this has also been requested by referee #2 and we will include another figure, showing peak sediment and peak water (see above). In the discussion (around line 553) we mention that conceptual models assume that the timing of peak sediment is partly determined by changes in erosive precipitation. "Indeed, for the study area of this study, a decrease in summer precipitation sums (i.e. June to August, which is the time of minimum snow-cover and thus maximum erodibility) is projected (Hanzer et al., 2018)[...]."*
*We added: "At the same time, heavy precipitation events are projected to become more intense (and only slightly more frequent, Fehler! Verweisquelle konnte nicht gefunden werden. in the appendix). However, the negative trend in discharge appears to prevail, as our estimates suggest that 'peak sediment' coincides with 'peak meltwater'."*

This study is the result of a series of studies to develop the model, test it and now apply it for CCI assessment. Generally, it seems the authors found a good balance between re-explaining the important parts and citing for more details. However, the exact inputs/features/predictors to the model remained unclear to me, as the plots only show T/P/Q but the text mentions also antecedent conditions. Clarifying this with e.g. a table would be important for the interpretation of the results.

*Answer: Thank you for pointing out that this was not clear. We have described this in more detail in the description (starting around line 170).*

I find the entire manuscript well written and structured. Except for section 3.3 (sensitivity/calssification), which I had to go back to several times. I guess you know this part is complicated so you came up with the conceptual figure 4 (which is nice). However, the use of max, percentiles, overlap, observation, etc. is confusing. Maybe there's no better way of denoting the variables but you could try to think of one. I also wonder if it wouldn't be easier to put the numbers of fig 5 into a bar plot or marking them in a time series somehow as a more intuitive way of looking at them. The box plots I find little informative.

*Answer: Thank you for this feedback. After putting quite some though into it, we did not find better names, but definitely see potential to improve the description of the sensitivity analysis in section 3.3. We simplified it to make sure it is more easily understandable. We understand and regret that fig 5 is not very intuitive. However, marking the numbers in a time series would not be straightforward, because each boxplot represents 3 – 14 projections of almost 100 years for each predictor, and anything below the maximum value in the training dataset is not of interest for the problem at hand. Bar plots in turn can only show the mean exceedance extent, or would also need to have whiskers to at least show min and max exceedance extent, and thus would not be much different from boxplots. We think that to show*

*the distribution of exceedance extents, boxplots are a good choice, as they show the mean as well as 25 and 75 percentile, and also outliers.*

**Specific comments**

L10: do you mean «downstream hydropower reservoirs»?
*__Answer:__ We agree that it makes sense to simplify this here and changed it.*

L11: do you mean "physical models" or "physically-based numerical models". Anyway, models exist but it can be difficult to calibrate/validate because observations are scarce. Furthermore, quantifying climate change impacts on hydro-geomorphic processes is subject to uncertainties because of low signal-to-noise ratios in climate and because such systems are non-linear (sediment storage, geomorphic thresholds, etc.)
*__Answer:__ Thank you for this attentive comment. Maybe process-based models is a better general term here. We will change this. Exactly, we mention the lack of sufficient observations that are available for model development around line 71, but tried to keep it (really) short in the abstract.*

L15-25 I think this method part is lengthy in an abstract. It has many details. I would leave out details like catchment area, and glacier area, specific dataset names, etc. Instead, you could briefly say what are the in- and outputs of your model, what you do with the output in one sentence and where you apply it to and then go on to the results
*__Answer:__ Thank you. We agree that glacier area is unnecessary here, but find names of datasets and the catchment area important (for example, a reader might wonder if it is comparable to their study area?). We streamlined this paragraph to shorten it.*

L23: I would write "projection period"
*__Answer:__ Thanks, we changed this throughout the manuscript.*

L37: "up to an order of magnitude"
*__Answer:__ Thank you, we changed this.*

L46: what do you mean by "those changes go hand in hand with changes in discharge…"? That they change proportionately/linearly or that climate change also affects those other things?
*__Answer:__ Thank you for pointing out that this was unclear. We mean that the changes in the cryosphere cause changes in discharge. We rephrased this.*

L44-L51: this paragraph is to show how CC affects hydrology, geomorphology, and how it's coupled, which is a good idea. However, how sediment transport is affected remains very superficially here. You could for example say that glacial retreat and permafrost are expected to increase sedment availability as it exposes glacial till and weakens rock walls, and I think some of your references do studies in that direction.
*__Answer:__ Thank you, we included this.*

L66: debris flows are also a massmovement
*__Answer:__ Thank you, we removed "debris flows".*

L79-91: as an addition to this nice list of models you should also consider recent work by Cache et al. https://doi.org/10.1016/j.geomorph.2023.108782. Not sure they considered glaciers, but snow.

*__Answer:__ Thank you for bringing this paper to our awareness, which is indeed very interesting! However, we do not think it fits to mention it in the above mentioned paragraph, where we attempt to give a brief overview of models that have been used for partially glaciated / deglaciating catchments, to explain the point that existing models cannot capture all relevant processes. We are aware of CAESAR-Lisflood, but to our knowledge, it is not applicable to glaciated catchments. The paper you mention also deals with a pre-Alpine but non-glaciated catchment, which thus functions in a fundamentally different way with respect to erosion processes.*

L99-101: why is that so?

*__Answer:__ Because such black box methods tend to perform well for black box problems such as high-alpine sediment dynamics, 'where the input data and output data are well-understood or at least fairly simple, yet the process that relates the input to output is extremely complex' (Lantz, 2019). We have added this to the paragraph.*

L113-115: to me it seems like you should change the order of i) and ii) as you need to assess uncertainties in order to study changes, i.e. their significance

*__Answer:__ Thank you, we changed this.*

L138-171: This section is very clear. I miss some information on how the QRF was validated and also on the temporal resolution of the model. Is the input based on daily data and then you assess it at the annual resolution? Additionally, a table with the clearly defined features (you mention antecedent conditions) fed to the QRF I think is important. I know it's in your previous papers but these predictors and their definition are crucial for your CCI assessment.

*__Answer:__ Thank you for pointing this out. We have provided more details on the features around line 170 (see answer above). We added a short description of the validation in the previous paper and the temporal resolution (daily) to the paragraph. Indeed, we trained and applied the model at daily resolution, and (mainly) analyze annual yields, but also the seasonality (i.e. at monthly resolution) and changes in the number of days with heavy precipitation (appendix).*

L173: as you mention the relevance of mass movements, do you know anthing about that in your study site?

*__Answer:__ Yes, there was a mass movement that covered parts of the tongue of the Hintereisferner, within the Vent catchment, in 2020, which is within our training data at gauge Vent. There was another period with very high SSY, which we assume was another mass movement in 2014, but we do not have direct observations of this. See also our paper here (DOI 10.5194/esurf-10-653-2022), where we discuss this in section 4.5. Additionally, there was an observed small debris flow in 2019 (also part of the training data at gauge Vent).*

L216-218: which stations (I assume a national network with 30+ years of observations) and how is the quantile mapping at the point scale translated to the grid?

*__Answer:__ With respect to the first part of the question, we moved the sentence from lines 238-240 (line numbers in the earlier version) to this paragraph. With respect to the second part of the question, quoting Hanzer et al., 2018: "With regard to the selection of RCM grid points for the downscaling to the point scale, we followed the approach by Hofer et al. (2017) (who however used linear regressions rather than QM) to find the optimum scale (OS) for each*

*station and target variable: for each station and variable, spatial averages of the closest 1 ×*
*1, 2 × 2, . . . , 10 × 10 RCM grid points were calculated and subsequently used for bias*
*correction. The OS for a given station and variable was then defined as the spatial window*
*which minimizes the deviations between the cumulative distribution functions of the corrected*
*and observed data in terms of the mean absolute error (MAE)." We think that this is too*
*detailed to include it in our manuscript – especially as the paper by Hanzer et al. with the*
*detailed information is openly available in HESS.*

L220-L221: why are only the s closest to the gauges considered? Their effect on the sediment
yield should be relatively small, as the sediment yield is a result of what is going on in the
catchment.
***Answer:*** *Thank you for this interesting question. We chose these points, because this setup is*
*most similar to the data that the QRF models had been trained on in the previous study, which*
*were meteo data recorded at or very close to the respective gauges. We did consider using*
*gridded data in the previous study as well, but firstly, the aim of that study was to be able to*
*estimate yields several decades into the past and gridded data are available only since 2003,*
*and secondly, these gridded data also bear some challenges and come with uncertainties*
*(such as potential severe underestimations of precipitation amounts in higher elevations, and*
*there are no stations within out catchments used to produce the gridded data). Since QRF is a*
*data-driven approach, we cannot feed the trained model with an entirely different type of data*
*(for example catchment sums of precipitation or catchment mean temperature) and expect*
*that the learned relationships still apply. As a minor technical point, the time series for the*
*points close to the gauges were the ones that were readily available from Hanzer et al.; in*
*order to acquire the gridded datasets, they would have had to re-run their models. Lastly,*
*since the catchments are not that big (100 and 11 km²), we'd expect that at least temperature*
*in the catchment will be highly correlated to temperature at the gauges, and few precipitation*
*events will occur only in the top part of the catchment and not at all at the gauges.*
*Additionally, some information on precipitation (at least the big events, which are likely most*
*relevant to erosion) will also be present in discharge.*

L238-240: I think this should go to the previous section (and partly answers my question
L216)
***Answer:*** *Thank you, we moved this sentence to the previous section.*

L273: QRF models or model?
***Answer:*** *modelS, because it is two separately trained models for the two gauges. Yet, we*
*understand the confusion, because we only mention one of the gauges in the same sentence.*
*We removed "of our QRF models" and changed it to "the obtained projections".*

L297: how was the maxima chosen? Is it a percentile or the actual max? how do you know
it's not just an outlier? Justifying this is quite important as it affects many results.
***Answer:*** *Thank you for this question. It is the actual max. We believe this makes sense,*
*because this addresses the technical limitation or QRF. Naturally, these data points will most*
*likely be outliers (depending on how you define outlier?), because such very high/extreme*
*values tend to be rare. However, we don't think that they are errors in the data (such as*
*measurement errors) because the observation data have been quality checked, e.g. by the*
*hydrographic service at gauge Vent.*

L317: missing word "in the …"
***Answer:*** *Thank you for pointing this out, we have corrected this.*

Eq. 3: "14 a" is a bit confusing as it's a mix of variables and quantities. I suggest changing "14 a" to t_ov or similar and specify in the text that it is 14 years. And what is rounded to whole days?
*__Answer:__ Thank you, we changed this. "Whole days" means whole numbers / integers (as opposed to e.g. 1.54 days). We corrected this.*

Figure 6: I cannot see the Amundsen data. X-axis label is missing.
*__Answer:__ Thank you, we added the x-axis label (month). The AMUNDSEN data are the solid (original) and dashed (bias-corrected) lines, colored by RCP. We improved the figure description to prevent confusion here.*

Table 5: if 2/3 does not oocur you can just leave it out
*__Answer:__ We politely disagree, because "more than 1/3" could also be > 2/3, but this way, the reader knows that everything that is more than 1/3 is also less than 2/3.*

L413: why are different variables driving the SS in these catchments?
*__Answer:__ Thank you for this comment. As we mentioned above, the sensitivity analysis does not necessarily indicate the driving predictors, but is meant to give information on the extent to which annual SSY estimates may be affected by underestimations on days with OOOR observations for the different predictors. We stated this more clearly here to avoid confusion.*

Figure 9: over which time period is this?
*__Answer:__ 2007 – 2020, see line 332 and after. We added this to the figure description and improved the description of the sensitivity analysis (see comment above).*

L513: where can we see peak water?
*__Answer:__ as suggested above, we added a new figure, showing peak sediment and peak water.*

L521: maybe it would be good to repeat here how you define erosive precip
*__Answer:__ We are citing Zhang et al. here, who did not specify this further... Yet we suggested to add a sentence on heavy precipitation events in an earlier comment, which we think addresses this concern as well.*

L521-525: since your model is data-driven and not physically-based, it is good that you back up this result with findings from literature. However, a hypothesis for the physical explanation of this result would be good.
*__Answer:__ we have addressed this in a previous comment (summer precip decreases; heavy precip events are projected to become more intense but only slightly more frequent; the negative trend in discharge appears to prevail; see above).*

L525: where do we see peak meltwater?
*__Answer:__ see above.*

L554-L556: since you mention the relevance of your work for the hydropower sector in the introduction, a statement to how this sector will be affected would fit here
*__Answer:__ thank you for pointing this out. We included it in the paragraph above, (since the increase in high-magnitude sediment events is likely more of a problem than a reduction in overall sediment transport for the hydropower sector).*

 L574: general comment: when you mention your results/figures in the discussion, you can also reference them

***Answer:*** *Thank you, we included references to the figures throughout the discussion where it was appropriate.*

---

## Author Response (AR2)

*Dear Manuela Brunner, dear anonymous referee,*

*We would like to thank you – again – for your detailed and constructive comments, questions and suggestions. Below, we provide our responses as direct answers to each comment and point out the changes made to the manuscript.*

*We hope that this will be to your satisfaction.*
*Best, Lena Katharina Schmidt on behalf of all authors*

I thank the authors for addressing most of my previous review comments. I think that the readers will now better understand what has been done in this study. My main critique this time is that uncertainty should be embraced more. More specifically, the first two subsections of the discussion on future changes is based on the median change. However, as seen in Figure 10, none of these changes seem to be significant and remain within the uncertainties. This is a common challenge in hydro-geomorphic climate change studies and should be properly acknowledged to avoid overinterpretation. However, if this point is addressed and the minor comments below, I think this study is ready for publication in HESS. I congratulate the authors on their work.

**_Answer:_** *Thank you for pointing out that we did not specify in the discussion, that we are referring to the ensemble mean of annual SSY of all models. We corrected this to avoid confusion. However, with respect to your point about changes not being significant: In section 3.4.1 we described that the trends in the ensemble means are negative and highly significant for all RCPs (see table 6), that trends in mean annual SSY of most individual models (26 of 31 at gauge Vent and 30 out of 31 at gauge Vernagt) are negative and significant, and that even the 99th percentiles show significant negative trends at both gauges (except for RCP4.5 at gauge Vent which is close to zero and not significant). We think this misunderstanding might result from the figure description of figure 10, so we adapted it to avoid confusion (it shows the mean of all models as a thick line and the min and max of the individual models for each year).*

L10: I would say either "yet" or "so far"
**_Answer:_** *thank you, we have changed this.*

L64: What do you mean by "capture"? Identifying, monitoring, modelling,…?
**_Answer:_** *What we mean is that it is difficult to capture in process-based models. We have changed it to "model".*

L159-163: This seems repetitive as there's a similar argument around L110 in the into. Please consider cutting.
**_Answer:_** *Thank you, we agree that it is similar to line 110. However, it is more detailed here (which ML approaches exactly etc.), and we believe it is an important point to make (i.e. why we chose this approach based on the existing literature) – especially since we expect that most readers will not actually read both the methods and the introduction in detail.*

L181: I may be wrong but isn't it problematic to have day of the year as input when doing climate change impacts? This makes sense for stationary climate, e.g. to account for seasonal hysteresis, but wouldn't it affect your results strangely when there are seasonal shifts in in other drivers (e.g. discharge)?

***Answer:*** *Thank you for this interesting question. We would argue that using DOY is necessary, because in this way the model can distinguish between e.g. the same volume of discharge in spring, summer and autumn (which will likely have different effects on sediment export), and likely not problematic since it is not the only predictor. However, we also mention around line 603 that if the relation between the predictors and sediment export shifts fundamentally (e.g. as glaciers disappear), the model cannot capture this as this effect is not present in the training data.*

L196: can you provide mean temperature too?
***Answer:*** *Thank you, we added this information.*

L220: for examples
***Answer:*** *Thank you, we changed this.*

L233: Please check waht in-text citation should look like. Ususally Hanzer et al. (2018). You use without year here and year within commas in L223
***Answer:*** *Thank you for pointing this out, we have corrected this.*

L237: I think "cannot" is more formal than "can not"
***Answer:*** *Thank you, we have corrected this.*

Figure 10: Please use labels a)-d), it's also easier to refer to in the text. Please also mention why the >2070 is faded
***Answer:*** *Thank you, we have added the labels and improved the figure description.*

Figure 11: Please also use labels and I think the title of the lower should be "Vent"
***Answer:*** *Thank you for pointing this out, we have corrected this and added labels.*

L509: what does 1/3 to ½ mean? Is this the uncertainty or just an visual estimate? Consider using calculated percentage.
***Answer:*** *Thank you for pointing out that this was confusing, we decided to eliminate that part of the sentence.*

L511: to me it still looks like onset of sediment export still is in May, although at a higher rate in future
***Answer:*** *Thank you, we have adapted the description.*

L516: I suggest putting the figure in the appendix instead of "not shown"
***Answer:*** *Thank you, we have added the figure to the appendix.*

Figure12: please specify in the caption that this if this is the mean of all climate-model chains
***Answer:*** *Thank you, we have specified this.*

L524: the discussion is missing subsection numbers
***Answer:*** *Thank you, we added them.*

L525-529: In my opinion the discussion doesn't need such a summary and rather fits the abstract or conclusion – up to you if you want to keep it.
***Answer:*** *Thank you, we would like to keep it since we do not expect most readers to read the entire*

*paper carefully, so that this short summary might be nice to have.*

L558-560: the conclusion of this is that the emissions are not the biggest uncertainty, but the internal climate variability is dominating. This is known to be important for climate modelling (https://doi.org/10.1007/s00382-010-0977-x), but has also been shown to be important for erosion/sediment modelling (https://doi.org/10.1007/s00382-010-0977-x or https://doi.org/10.1029/2020JF005739). I suggest you cite one of those.
**_Answer:_** *Thank you for this important comment, we have cited the two papers (two of the links point to the same paper).*

L668: please specify the influence of what is negligible
**_Answer:_** *Thank you, we have specified this.*

L570-671: maybe "hydro-geomorphic" instead of "hydro-sedimentological"?
**_Answer:_** *Thank you, we have changed this.*

---

## Author Response (AR3)

*Dear Manuela Brunner*

*Thank you very much for the timely handling of our paper.*
*We have addressed your technical remarks: we included a footnote to explain ibid. (ibidem (latin): in the same place; used in referring again to the book, page, etc. cited just before) and have provided the figures in higher resolution.*

*We hope that this will be to your satisfaction.*
*Best, Lena Katharina Schmidt on behalf of all authors*